# Toll-1-dependent immune evasion induced by fungal infection leads to cell loss in the *Drosophila* brain

Deepanshu N. D. Singh[1,2¤a], Abigail R. E. Roberts[1¤b], Xiaocui Wang[1], Guiyi Li[1], Enrique Quesada Moraga[3], David Alliband[1¤c], Elizabeth Ballou[2¤d], Hung-Ji Tsai[2], Alicia Hidalgo[1]*

**1** Brain Plasticity & Regeneration Lab, Birmingham Centre for Neurogenetics, School of Biosciences, University of Birmingham, Birmingham, United Kingdom, **2** Institute of Immunity and Infection, School of Biosciences, University of Birmingham, Birmingham, United Kingdom, **3** Departamento de Agronomía, Universidad de Córdoba, ETSIAM, Córdoba, Spain

¤aCurrent address: Soller Lab, University of Manchester, Manchester, United Kingdom
¤bCurrent address: School of Mathematical Sciences, University of Sheffield, Sheffield, United Kingdom
¤cCurrent address: University of Bristol, Bristol, United Kingdom
¤dCurrent address: MRC Centre for Medical Mycology, Geoffrey Pope Building, University of Exeter, Exeter, United Kingdom

* a.hidalgo@bham.ac.uk

## Abstract

Fungi can intervene in hosts' brain function. In humans, they can drive neuroinflammation, neurodegenerative diseases and psychiatric disorders. However, how fungi alter the host brain is unknown. The mechanism underlying innate immunity to fungi is well-known and universally conserved downstream of shared Toll/TLR receptors, which via the adaptor MyD88 and the transcription factor Dif/NFκB, induce the expression of antimicrobial peptides (AMPs). However, in the brain, Toll-1 could also drive an alternative pathway via Sarm, which causes cell death instead. Sarm is the universal inhibitor of MyD88 and could drive immune evasion. Here, we show that exposure to the fungus *Beauveria bassiana* reduced fly life span, impaired locomotion and caused neurodegeneration. *Beauveria bassiana* entered the *Drosophila* brain and induced the up-regulation of *AMPs*, and the Toll adaptors *wek* and *sarm*, within the brain. RNAi knockdown of *Toll-1, wek* or *sarm* concomitantly with infection prevented *B. bassiana*-induced cell loss. By contrast, over-expression of *wek* or *sarm* was sufficient to cause neuronal loss in the absence of infection. Thus, *B. bassiana* caused cell loss in the host brain via Toll-1/Wek/Sarm signalling driving immune evasion. A similar activation of Sarm downstream of TLRs upon fungal infections could underlie psychiatric and neurodegenerative diseases in humans.

## Introduction

Fungi can manipulate the behaviour of their insect hosts and induce neurodegeneration, ultimately causing host death and enabling fungal growth and reproduction [1–3]. How fungi can overcome protective brain barriers is unknown. Even exposure to fungal volatiles alone is sufficient to reduce life span, decrease dopamine levels, induce neurodegeneration and impair

**Data availability statement:** All relevant data are within the paper and its Supporting Information files. ImageJ plugins used in this study are available from the Institutional Data Access repository UBIRA https://edata.bham. ac.uk/ "Software plugins for automatic cell counting in the Drosophila central nervous system" DOI: https://doi.org/10.25500/edata. bham.00001213.

**Funding:** This work was funded by a Darwin Trust Doctoral Studentship to DNDS, and BBSRC Project Grant BB/R017034/1 and Wellcome Trust Investigator Award 223197/Z/21/Z to AH. The funders had no role in study design, data collection and analysis, decision to publish, or preparation of the manuscript.

**Competing interests:** The authors have declared that no competing interests exist.

**Abbreviations:** AMP, antimicrobial peptide; BBB, blood-brain barrier; CLW, calcofluor white; DAN, dopaminergic neurons; GNBP3, Gram Negative Bacteria Binding Protein 3; OD, optical density; PBS, phosphate buffer saline; PER, proboscis extension response; PFA, paraformaldehyde; PI, performance index; PRR, pattern recognition receptors; SDA, sabouraud dextrose agar; SING, Startled-induced negative geotaxis; SPE, Spätzle Processing Enzyme; TH, tyrosine hydroxylase; TLRs, Toll-like receptors.

locomotion in fruit-flies [4,5]. This is concerning also for human health. Abundant fungal spores, mould and fungal volatiles are commonly found in indoor damp conditions [6,7]. Fungal spores have been found in the brains of Parkinson's and Alzheimer's disease patients, and fungal infections are emerging as drivers of neuroinflammation, neurodegenerative diseases and psychiatric disorders [8–14]. How fungi and neuroinflammation drive disease in the host brain is unknown [8,11].

The molecular mechanism underlying innate immunity in response to fungal infections is universally conserved and well known. It was originally discovered in the fruit-fly *Drosophila melanogaster*, where fungi lead to the activation of the Toll receptor [15–17]. This led to the discovery of Toll homologues in mammals, the Toll-like receptors (TLRs), and Tolls and TLRs across organisms have a universally conserved function in innate immunity [17–20]. Mammalian TLRs are pattern recognition receptors (PRR) that directly bind pathogens [20], but in *Drosophila* fungi are recognised by the PRR Gram Negative Bacteria Binding Protein 3 (GNBP3) instead [21]. This initiates a proteolytic cascade that activates Spätzle Processing Enzyme (SPE) that cleaves Spätzle (Spz), ligand of Toll-1 [20,22–25]. Signalling downstream of Toll-1 and TLRs proceeds via MyD88, triggering the nuclear translocation of the transcription factor Dif/NF-κB, which activates the expression of antimicrobial peptide (AMP) encoding genes [15,16,26,27]. In fruit-flies, AMPs include *drosomycin (drs)* and *metchnikowin (mtk),* which protect *Drosophila* from fungal infections [16,28–32].

However, in the nervous system, Toll receptors can also signal via the alternative adaptors, Weckle (Wek) and Sterile-α and Armadillo Motif containing protein (Sarm) [33,34]. Toll signalling can promote either cell survival via Wek-MyD88-NFκB or cell death via Wek-Sarm-JNK signalling downstream [34–36]. Tolls can also promote cell proliferation or quiescence, depending on cell context [34–36]. Sarm is expressed in neurons, and it is the universal, highly conserved inhibitor of MyD88 and TRIF signalling [26,37–40]. Sarm can be at the plasma membrane and associated with mitochondria, it can induce neuronal death via inhibiting the pro-survival function of MyD88, via activating JNK signalling and via NADase catalytic activity, which also drives neurite self-destruction [26,34,41–48].

There are nine *Toll* paralogues in *Drosophila*, and at least seven are expressed in the adult fly brain [34–36]. In the nervous system, Tolls have functions unrelated to immunity, in axonal connectivity, neurogenesis, cell survival, cell death, structural brain and synaptic plasticity [34–36,49–51]. Not all Tolls function equally, and Toll-1 is more likely to drive apoptosis than others [34–36]. Importantly, Toll-1 has the most prominent function in innate immunity [16,52,53].

Here, we asked whether fungal infections could induce neurodegeneration in the host brain via Toll-Wek-Sarm signalling. We used *Beauveria bassiana,* an entomopathogenic fungus that induces disease in over 700 arthropod species, a plant endophyte that protects plants against insects, can be used for pest control, and is a well-known activator of Toll signalling in *Drosophila* [21,54–62]. The upstream route to the activation of Toll-1 from GNBP3 to cleavage of Spz and activation of Toll signalling is very well known [15–17], thus was not explored further here. Instead, we asked: (1) whether *B. bassiana* could enter the brain and (2) could activate the alternative immune evasion Sarm pathway. We show that the fungus benefits from immune receptor signaling to cause cell death in the host brain.

## Results

### *B. bassiana* impaired behaviour and entered the brain

It has previously been reported that exposure to fungal volatiles is sufficient to reduce life span and climbing and induce neurodegeneration in *Drosophila* [4,5]. To ask whether and

how the fungus *B. bassiana* – which is well-known to activate Toll signalling – could affect the brain, we mimicked natural exposure of flies using an infection chamber (Fig 1A). Here, a *B. bassiana* (strain 80.2) fungal culture was established in a fly bottle, flies were inserted and shaken to get them directly exposed to spores, and thereafter kept there for the duration of the experiments. Wild-type non-infected control flies lived up to 70 days, but flies exposed to *B. bassiana* died within less than 20 days and by day seven more than half of the flies had died (Fig 1B). Negative geotaxis—also known as startle-induced negative geotaxis (SING) or the climbing assay—is commonly used to measure locomotor impairment caused by neurodegeneration [63,64]. No effect was seen after exposure for three days to *B. bassiana*, but seven days of exposure impaired climbing (Fig 1C). Altogether, these data showed that exposure to *B. bassiana* decreased locomotion and longevity. Multiple factors could directly and indirectly affect both locomotion and survival in diseased flies after infection. Nevertheless, these phenotypes are also common indicators of neurodegeneration in flies.

We wondered whether *B. bassiana* could affect brain function by entering the brain. After three days exposure of wild-type flies to *B. bassiana*, fungal cells were detected within the brain using the FM4–64 dye together with nuclear DAPI (Figs 2A and S1A). A second dye, calcofluor white (CLW) also revealed fungal cells in the optic lobes and central brain (Fig 2B). Finally, adult wild-type flies were exposed to a GFP-transgenic *B. bassiana* strain (EABb 04/01-Tip GFP5) [65], and abundant GFP + cells were found within the optic lobes and central brain (Figs 2C-F, S1B and S1C). Interestingly, *B. bassiana-GFP* cells were found at the entry point of the proboscis into the brain (Fig 2C and 2E) and at the exit point of the esophagus (Fig 2E). *B. bassiana-GFP* cells were also found across the blood-brain barrier (BBB) (Fig 2D), which is formed of glial cells in *Drosophila.* Furthermore, germlings or germinating spores, with clear filaments or germtubes were also abundant in the brain (Fig 2F-F"). These findings show that *B. bassiana* infiltrated the adult brain. To enter the brain, *B. bassiana* would either have to damage the BBB or be carried across by macrophages [66]. The dye Dextran Red is commonly used to test the integrity of the BBB in *Drosophila* [67]. Following an injection into the thorax, Dextran Red outlines the retina's edge if the BBB is intact, whereas if it is not, the dye leaks into the retina [67]. We found that in flies that had been exposed to *B.*

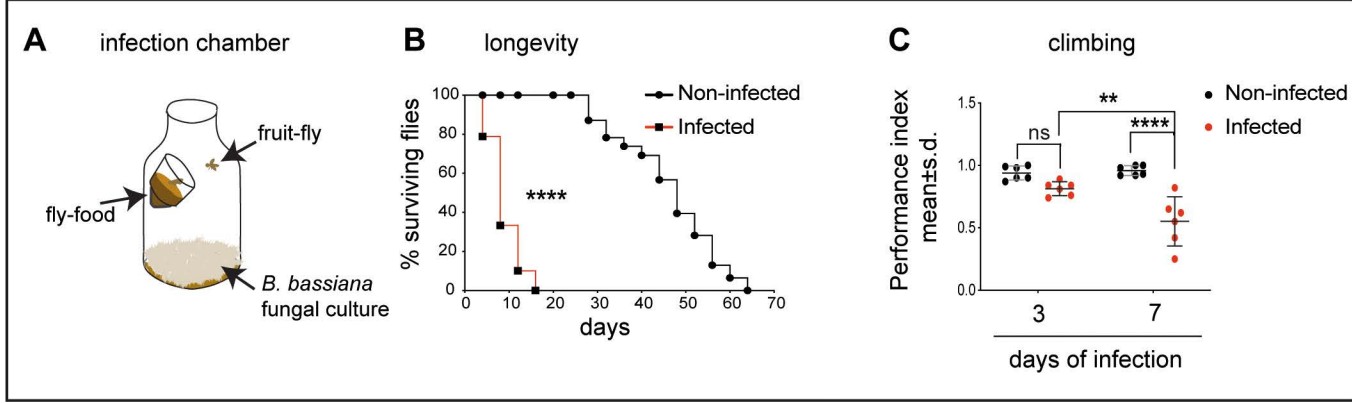

**Fig 1. *B. bassiana* decreased longevity and impaired climbing. (A)** Infection chamber. **(B)** *B. bassiana* exposure reduced life span of wild-type (*Oregon/CantonS*) flies. Percentage surviving flies at each scored day Log-rank test *p* < 0.0001, *n* = 104 non-infected flies, and infected flies *n* = 104. **(C)** *B. bassiana* exposure impaired climbing of wild-type (*Oregon/CantonS*) flies, after 7 days of exposure. Dot plots, with lines indicating the mean and standard deviation. Each dot represents one cohort of 7–10 flies. Two-way ANOVA group statistics: 3-days-after infection versus 7-days-after infection (Row factor) *p* = 0.0122; Noninfected versus infected (Column factor): *p* < 0.0001; Interaction: *p* = 0.0048. Asterisks in the figure show Turkey's multiple comparison correction tests. Sample sizes: 3 days after infection: n = 60 for each infected and non-infected. 7 days after infection: n = 51 for each infected and non-infected. **\*\****p* < 0.01, \*\*\**p* < 0.001. The data underlying panels B and C can be found in S5 Table Source data.

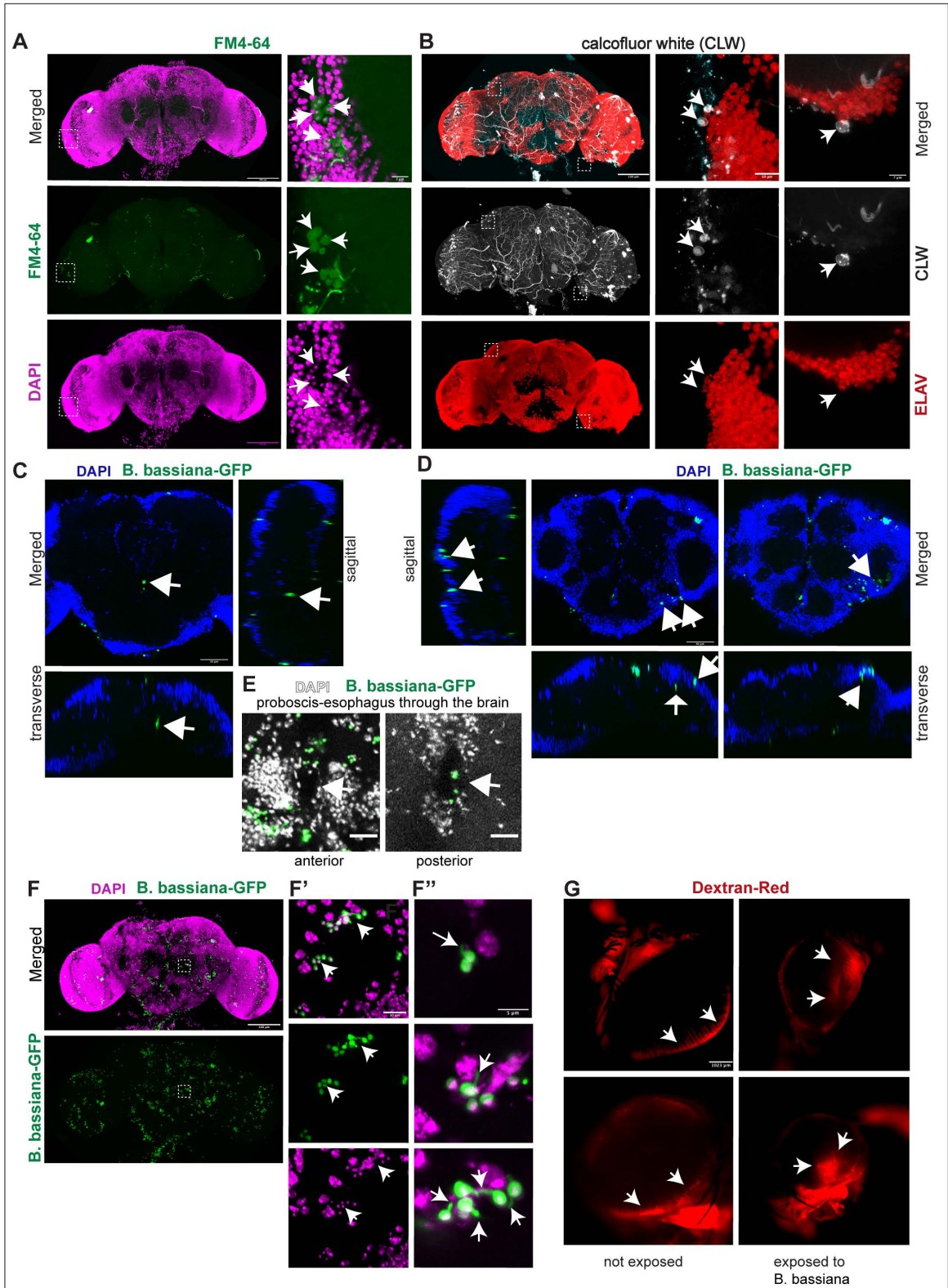

**Fig 2. *B. bassiana* was detected within the fly brain.** *B. bassiana* infiltrated the *Drosophila* adult brain of wild-type flies (*Oregon R*) as visualised with **(A)** FM4–64 dye, revealing fungal cells as they co-localise with the nuclear dye DAPI+ (arrows). **(B)** Calcofluor white, revealing fungal cells adjacent to neurons within the central brain (arrows). **(C)** Transgenic *B. bassiana*-GFP found at the

point of entry of the proboscis into the brain (arrows). Orthogonal views of one optical section. Nuclei labelled with DAPI. **(D)** DAPI labelled nuclei reveal the outer surface of the brain, including the BBB and *B. bassiana*-GFP crossing the BBB (sagittal and transverse views) and within the brain (transverse view). Orthogonal views of one optical section. **(E)** *B. bassiana-GFP* found at the entry point of the proboscis into the brain (arrow, anterior brain) and exit point of the oesophagus from the brain (arrow, posterior brain). **(F-F")** *B. bassiana*-GFP germlings or germinating spores colocalise with the neuronal marker DAPI (arrows, **F'**) and have filaments (arrows, **F"**). **(G)** *B. bassiana* damaged the blood-brain barrier in the eye, as Dextran Red could diffuse within the retina in infected brains (arrows) but not in non-infected controls. Scale bars: (A, B, F) left: 100 mm; (A, B) right: 7 mm; (C, D) 50 μm; (E) 20 μm; (F'): 10 μm; (F"): 5 μm; (G) 1,023 μm. Sample sizes: (1) FM4–64: *n* = 4/10 infected brains had fungal cells in the brain; non-infected n = 7. (2) CLW: *n* = 2/2 brains had fungal cells in the brain. (3) *B. bassiana-GFP*: *n* = 3/3 infected brains had spores in the brain; non-infected controls: *n* = 3. Non-infected controls for (A, C, D, E) in S1 Fig. Genotype of *Drosophila melanogaster* for all images: wild-type Oregon. Genotype of *B. bassiana*: 80.2 (A, B and F) and B. *bassiana-GFP*: EA-Bb-Tip 04/01 (C–E). See S1 Table for statistical details.

*bassiana* for seven days, Dextran Red spread within the retina (Fig 2G), meaning that the BBB was damaged. Altogether, these findings showed that following exposure, *B. bassiana* penetrated the fruit-fly brain.

## Exposure to *B. bassiana* activated Toll signalling within the adult brain

Fungi can degrade insects' cuticle, enabling them to penetrate hosts' bodies [68–71]. In this way, *B. bassiana* could invade the brain through the optic lobes, the head capsule and the proboscis. The proboscis is used for feeding [72], and it traverses the brain to join the oesophagus, thus providing a route for *B. bassiana* into the brain. However, *B. bassiana* could trigger avoidance in the fly, like in *C. elegans* worms, where pathogenic bacteria induce the worms to turn away [73,74]. Thus, we tested whether flies feed on *B. bassiana* spores. We offered them water or sucrose or spores mixed with blue dye and measured optical density (OD) in their abdomens after feeding. Flies fed on sucrose—which they are known to like—and similarly fed on spores, more than on water (Fig 3A). Thus, ingestion would provide *B. bassiana* with an effective route into the brain. To further test this, we next fed flies that had not been previously exposed to the fungus, with water containing *B. bassiana*-GFP spores. Following feeding, we detected *B. bassiana*-GFP in the brain (Fig 3B), in the proboscis (Fig 3B'), germlings (Fig 3B"), *B. bassiana*-GFP at the point where the proboscis enters the brain (Fig 3B''' anterior), at the point where the esophagus and neuropiles exit the brain (Fig 3B''' posterior) and over glial membranes deep within the brain (Fig 3B""). As these flies had not been exposed to *B. bassiana*-GFP other than through feeding, this means that ingested *B. bassiana* spores could penetrate the brain.

The findings that *B. bassiana* entered the brain and flies fed on it were surprising. Firstly, in *Drosophila*, the innate immune response driven by Toll-1 should clear the fungus [15]. Secondly, they contrast with the bacterial avoidance response of *C. elegans,* which depends on Tol-1 [73,74]. We reasoned that perhaps the proboscis might not express *Toll-1*. Thus, we asked whether *Toll-1* and *sarm* might be expressed in the proboscis. Using the in vivo reporters *Toll-1 > FlyBow* and *sarm^{NP0257} > FlyBow*, we found expression of *Toll-1* and *sarm* in sensory neurons of the proboscis (Fig 3C). We tested the consequences activating Sarm+ neurons might have on the proboscis extension response (PER), which is required for feeding [72]. Using two distinct *sarmGAL4* drivers, we found that activating Sarm neurons with TrpA1 increased the incidence of PER events compared to unstimulated controls (Fig 3D and 3E). This showed that *sarm*-expressing neurons are not involved in avoidance of *B. bassiana*, and instead could be involved in feeding.

Next, we wondered whether *B. bassiana* could activate Toll signalling inside the brain. All the Toll-1 signalling components – including *GNBP3*, the downstream *SPE* protease, *spz-1*, *Toll-1* plus six other *Toll* receptors, *MyD88*, *sarm*, *Dif/NF-κB* and target antimicrobial peptide

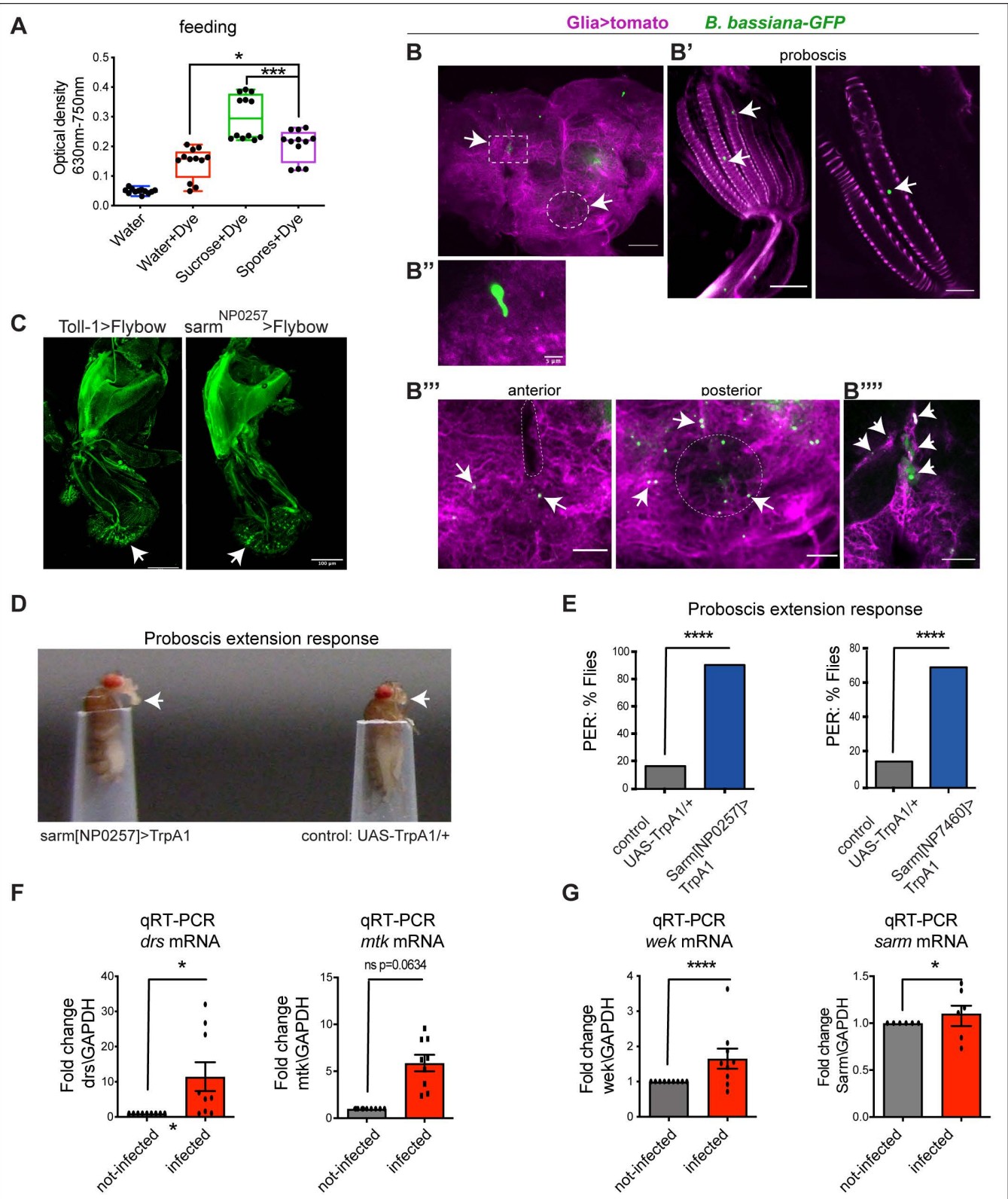

**Fig 3. Exposure to *B. bassiana* activated Toll signalling within the adult brain.** **(A)** *Drosophila* wild-type (*Oregon R*) flies fed on *B. bassiana* spores that had been mixed with blue dye diluted in water. One Way ANOVA *P* value < 0.0001. **(B)** Non-infected *repo>myr-tomato* flies were fed a solution of *B. bassiana-GFP* spores, after which GFP + cells were found: **(B)** in the central brain, **(B')** in the proboscis, **(B")** germling, **(B''')** in the entry site of the proboscis

(anterior) into the brain, and exit site of esophagus (posterior), and **(B"")** on glia within the brain. **(C)** *Toll-1 > FlyBow* and *sarm*NP0257 > *FlyBow* reveal expression in the proboscis, note axons and dendrites of sensory neurons in the labellum (arrows). *n* = 3–7 brains. **(D)** Activating Sarm+ neurons triggered the proboscis extension response, a proxy for feeding. **(E)** Quantification of **(D)**. Chi-square test, ****$p < 0.0001$. Test: *sarm*NP0257>*TrpA1* n = 34 flies, Control was responder line crossed to wild-type Oregon: *UAS-TrpA1/+* n=34 Test: *sarm*NP7460>*TrpA1* n = 40, Control: *UAS-TrpA1/+* n = 40. 40. **(F, G)** *B. bassiana*B. bassiana infection of wild-type (Oregon R) flies for 7 days raised the expression levels of *AMPs, wek* and *sarm* within the brain. qRT-PCR data comparing fold change levels of **(F)** *drs* and *mtk* mRNA and **(G)** *wek* and *sarm* in infected brains, normalized to GAPDH as a housekeeping control. DeltaCT mean ± standard deviation. *drs:* Unpaired Student *t* test, *$p < 0.05$, n = 3 biological replicates (b.r.); *mtk:* $p = 0.0634$; n = 3 b.r.; *wek:* Unpaired Student *t* test ****$p < 0.0001$ n = 4 b.r.; *sarm:* t test: *$p < 0.05$, n = 4 b.r. Scale bar: (B) 50 µm; (B', B", B"") 20 µm; (B") 5 µm; (C) 100 µm. ">" stands for GAL4/UAS and "+" stands for wild-type chromosome. See S4 Table for further details. The data underlying panel A, E, F and G can be found in S5 Table Source data.

genes *drs* and *mtk* – are normally expressed in the adult brain (Scope Fly Atlas) [36,75]. We visualised the expression of *Toll-1, MyD88* and *sarm* in the brain, showing they are widely expressed (S2A Fig). Using the DenMark reporter which labels cell membranes, MyD88 revealed the BBB ensheathing the brain (S2B and S2B' Fig). At least some of the MyD88 + cells at the BBB were glia, as when labelled with nuclear Histone-YFP, they co-localised with the nuclear glial marker Repo (S2C and S2C' Fig). Consistent with this, we had previously shown that MyD88+ cells include both neurons and glia [36]. Importantly, *Toll-1, MyD88* and *sarm* are all expressed in cells that surround the entry (anterior brain) and exit (posterior brain) sites of the proboscis and oesophagus, respectively, through the brain (S2D Fig). At least some of these MyD88 + cells surrounding the tunnel across the brain are glia, as they colocalise MyD88 > his-YFP and Repo (S2E Fig). This means that as *B. bassiana* travelled from the proboscis into the brain, it would encounter abundant cells expressing *Toll-1* and its signalling machinery.

Thus, next we asked whether exposure of wild-type flies to *B. bassiana* might activate innate immunity in the brain. At seven days post-infection, expression of AMPs *drs* mRNA was upregulated within the brain, and that of *mtk* also, albeit not significantly (Fig 3F). The mechanism of activation of Toll-1 is very well known. Instead, we wondered whether *B. bassiana* could also drive an immune evasion pathway downstream of Toll-1, via Wek with Sarm. Intriguingly, following infection, the expression of both *wek* and *sarm* was also upregulated in the brain (Fig 3G).

To conclude, *B. bassiana* could enter the brain also through feeding, and it induced Toll signalling within the adult brain, also engaging Wek and Sarm. Wek is not required for innate immunity [33], and it links Tolls to Sarm instead [34]. Sarm inhibits MyD88 and immune signalling. Toll signalling via Wek and Sarm induces apoptosis downstream [34], and Sarm also induces axonal destruction [46]. Thus, the infection-dependent up-regulation of *wek* and *sarm* downstream of Toll-1 could potentially drive neurodegeneration.

## Exposure to *B. bassiana* caused neuronal and glial loss

To ask whether *B. bassiana* could induce neurodegeneration via Toll-1, Wek and Sarm, we first tested whether exposure to *B. bassiana* caused cell loss in the brain. We used the nuclear reporter histone-YFP to visualize Sarm + cells (*sarm*$^{NP0257}$ > *hisYFP*) and quantified them automatically. Seven days exposure to *B. bassiana* decreased Sarm + cell number in the central brain (Fig 4A and 4B). As glial cells form the BBB in *Drosophila*, and at last, some of them are MyD88 + (S2B-E Fig) [36], we asked whether exposure to *B. bassiana* could affect glia. Glial cells were labelled with pan-glial anti-Repo antibodies, glial cell number in the brain was quantified automatically, and it also decreased with infection (Fig 4C and 4D). Importantly, loss of glial cells could lead to BBB breakdown and facilitate entry of *B. bassiana* into the brain. Exposure of fruit-flies to the fungal volatile 1-octen-3-ol caused loss of dopaminergic neurons [4]. Thus, we asked whether exposure to *B. bassiana* could elicit similar effects. We

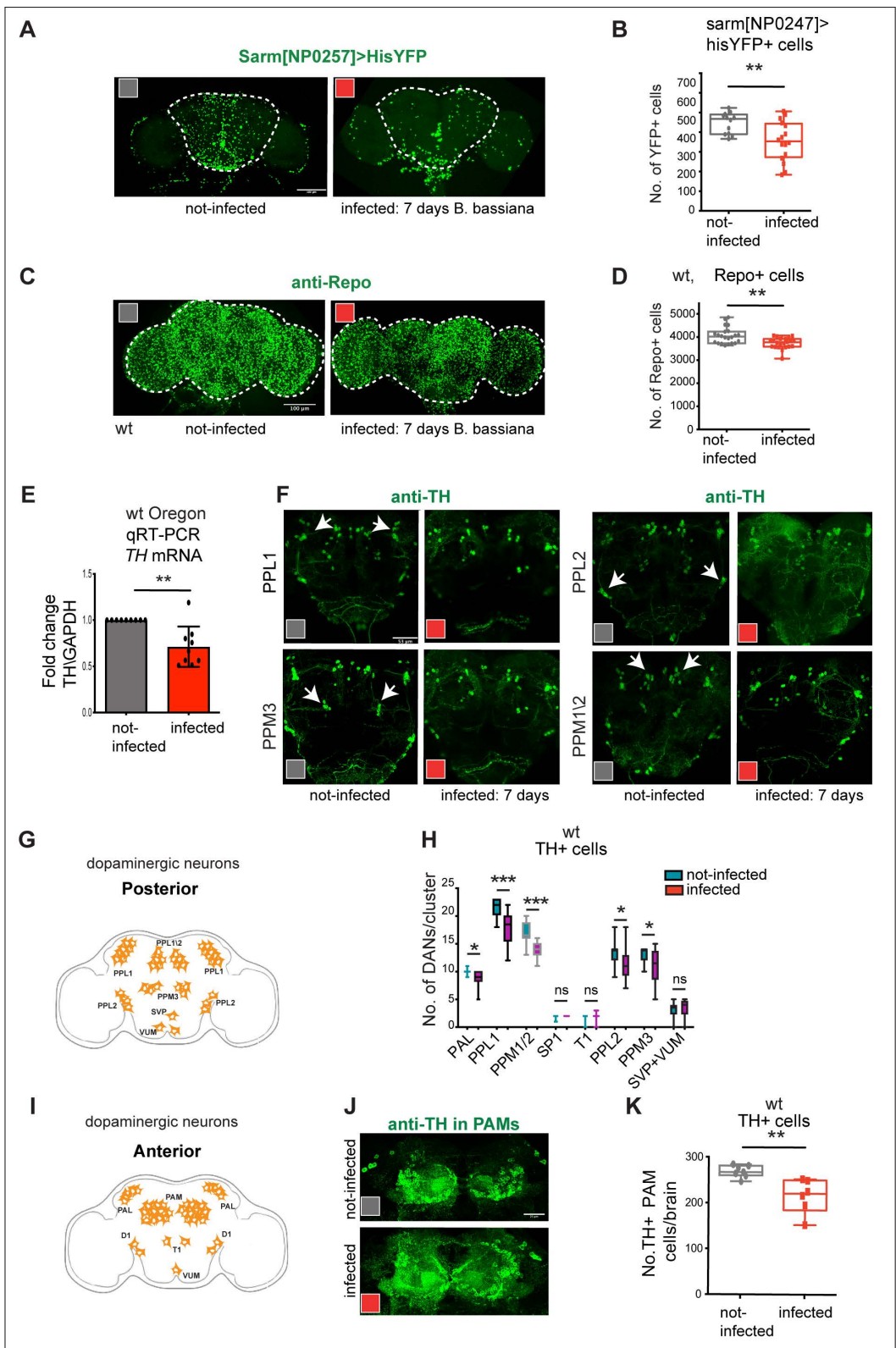

**Fig 4. Exposure to *B. bassiana* caused cell loss in the fly brain. (A,B)** Seven-day exposure to the *B. bassiana* caused loss of *sarm*NP0257 > *HisYFP* cells. Quantification in **(B)**, Mann-Whitney U test, *p* = 0.002. **(C,D)** Seven-day exposure to *B. bassiana* caused loss of glia cells, visualised with anti-Repo antibodies. Quantification in **(D)**, Unpaired Student *t*

test, $p = 0.002$. **(E)** Seven-day exposure to *B. bassiana* reduced *tyrosine hydroxylase* (*TH*) expression. qRT-PCR showing fold-change relative to non-infected wild-type controls and normalised to GAPDH, mean ± standard deviation. Unpaired Student *t* test on delta-Ct values $p = 0.0011$, $n = 3$ biological replicates. **(F)** Seven-day exposure to *B. bassiana* caused loss of dopaminergic neurons (DANs) visualised with anti-TH antibodies, in the posterior brain, quantification in **(H)**: PAL, SP1, T1: Mann-Whitney U tests; PPL1, PPM1/2, PPL2, PPM3, SVP, VUM: Student *t* test; non-infected brains $n = 12$, infected brains $n = 12$. Arrows point to each DAN class. **(G,I)** Illustration of dopaminergic neurons in the adult brain. **(J,K)** Seven-day exposure to the *B. bassiana* caused loss of PAMs, quantification in **(K)**: unpaired Student *t* test, $p = 0.0043$, non-infected brains $n = 7$, infected brains $n = 6$. Graphs in panels (B,D,K) show box-plots around the median, box with 50% of data points and whiskers with 25% of data points. Graphs in panels (E,H) show bar charts with mean ± standard deviation. Dotted lines in (A,C) indicate ROI for automatic cell counting with DeadEasy software. **$p < 0.01$, ****$p < 0.0001$. Scale bars: (A, C, F): 100 μm. (J) 25 μm. Genotypes for tests: (A, B) Infected flies bearing the Histone-YFP reporter expressed in Sarm + cells (*SarmNP0247 > hisYFP*); (C-F, H, J): wild-type Oregon R. Controls: non-infected flies of the same genotypes. See S4 Table for further details. The data underlying panels B, D, E, H and K can be found in S5 Table for source data.

first tested whether seven-day exposure to *B. bassiana* might affect the expression of tyrosine hydroxylase (TH), which is required for dopamine synthesis [76]. In fact, *B. bassiana* exposure caused a decrease in *TH* mRNA levels within adult brains (Fig 4E), meaning that *B. bassiana* exposure decreased dopamine production. Furthermore, using anti-TH antibodies, we found that at seven days post-exposure, the number of PPL1, PPL2, PPM1/2, PPM3 (Fig 4F-I), and PAM (Fig 4I, 4J and 4K) dopaminergic neurons had decreased.

Altogether, seven days exposure to *B. bassiana* caused loss of Sarm>his-YFP+ neurons, Repo + glia and TH + dopaminergic neurons in the infected adult fly brain.

## *B. bassiana* requires Toll-1 to induce cell loss in the fly brain

Toll-1 signalling can induce cell death via the Sarm/JNK pathway [34], so we next asked whether *B. bassiana*-induced cell loss depended on Toll-1. Toll-1 activates innate immunity through the well-known MyD88 pathway, which leads to the activation of Dif/NFκB and the expression of anti-microbial peptides (e.g., *drs*, *mtk*) [77]. *Toll-1, MyD88* and *sarm* are widely expressed in the brain, and at least *MyD88* also including in the BBB [36] (S2 Fig). Sarm is the inhibitor of MyD88 [34,39]. So in the brain, in MyD88+ cells, Toll-1 could potentially drive signalling via the two alternative MyD88 and Sarm pathways, in co-expressing cells. Thus, to account for both signalling routes, we visualised adult MyD88+ cells with *MyD88 > HisYFP* and counted cell number automatically. Seven-day exposure to *B. bassiana* caused MyD88+ cell loss (Fig 5A and 5C). To knock-down *Toll-1* expression specifically in adult flies, we used the temperature-sensitive Gal4 repressor tubulin-Gal80ts (Fig 5B, *tubGAL80ts, MyD88 > hisYFP*). In non-infected controls, adult *Toll-1* RNAi knockdown using line *UAS-Toll-1RNAikk/100078* (shown to work effectively in [36]) caused an increase in MyD88-YFP + cells within the central brain (Fig 5A-C). This increase in cell number could correspond to induced proliferation or increased cell survival. In fact, Toll-1 can induce apoptosis in at least larvae and pupae [34]. There, *Toll-1* loss of function increased neuronal number, and constitutively active *Toll-1¹⁰ᵇ* decreased neuron number and increased apoptosis [34]. This suggests that the increase in cell number caused by *Toll-1-RNAi* knock-down could result from decreased cell death. Importantly, in infected adult brains, *Toll-1 RNAi* knockdown prevented loss of MyD88-YFP + cells that would have been caused by *B. bassiana* exposure (Fig 5A-C). These data mean that *B. bassiana*-induced cell loss depends on Toll-1 signalling.

Neuronal loss can cause glial loss [78], *MyD88* is also expressed in Repo+ glial cells (S2B-D Fig) [36] and *Toll-1* is also expressed in glial cells [75]. Thus, we tested the effect on glial cell number. *Toll-1-RNAi* knock-down did not affect the number of Repo+ glial cells in non-infected controls (Fig 5D and 5E). However, *Toll-1* RNAi knock-down in MyD88+ cells prevented the decrease in glial cell number caused by *B. bassiana* infection (Fig 5D and 5E). Thus, *B. bassiana*-induced glial

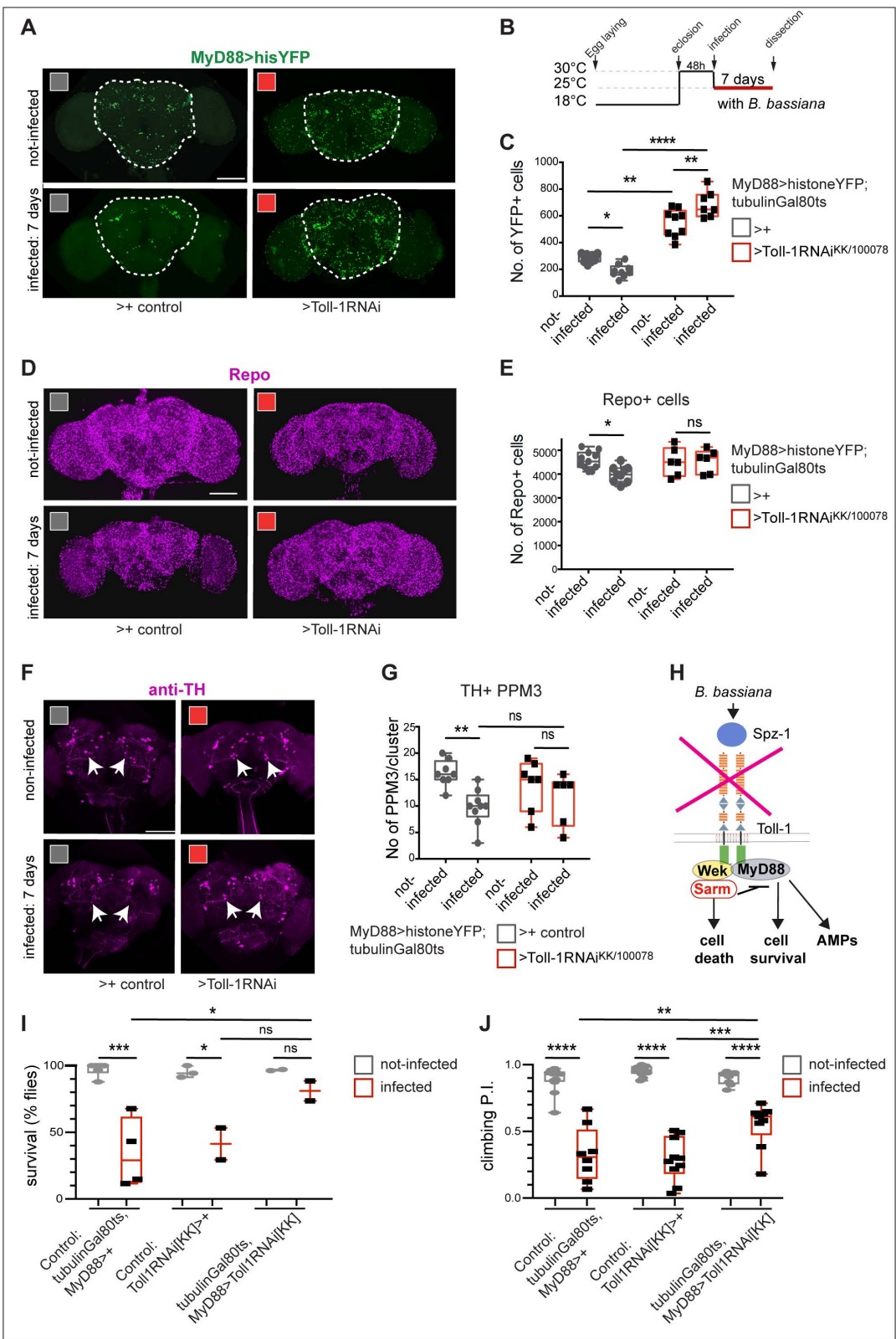

**Fig 5. *B. bassiana*-induced cell loss depends on Toll-1.** **(A)** Seven-day exposure to *B. bassiana* caused loss of MyD88+ cells, visualised with *tubGAL80*ts; *MyD88 > HisYFP*, and this was rescued with adult-specific *Toll-1-RNAi* [KK/100078] knockdown**. (B)** Diagram explaining the experimental temperature regime. **(C)** Quantification of data in **(A)**. Two-way ANOVA: Infected

versus not-infected: $p = 0.7557$; Genotypes: $p < 0.0001$; Interaction: $p < 0.0001$. Tukey's multiple comparison correction tests. Sample sizes: non-infected control brains $n = 10$, infected control brains $n = 9$, non-infected *Toll-1-RNAi* brains $n = 9$, infected *Toll-1RNAi* brains $n = 9$. **(D, E)** Seven-day exposure to *B. bassiana* caused loss of glial cells visualised with anti-Repo antibodies **(D)**, which was rescued with adult-specific *Toll-1-RNAi* KK/100078 knockdown, automatic quantification with DeadEasy in **(E)**. Two-way ANOVA: Infected versus not-infected: $p = 0.1003$; Genotypes: $p = 0.1285$; interaction: $p = 0.0666$. Tukey's multiple comparisons correction test. Sample sizes: non-infected control brains $n = 11$, infected control brains $n = 8$, non-infected Toll-1-RNAi brains $n = 6$, infected Toll-1-RNAi brains $n = 6$. **(F,G)** Seven-day exposure to *B. bassiana* caused loss of PPM3 DANs (arrows), visualised with anti-TH antibodies and this phenotype was rescued with adult-specific *Toll-1-RNAi* KK/100,078 knockdown. Arrows point to PPM3 neurons. Quantification in **(G)**. Two-way ANOVA: Infected versus not-infected: $p = 0.0042$; Control versus Toll-1 RNAi: $p = 0.8548$; interaction: $p = 0.1760$, and Tukey's multiple comparisons correction test. Sample sizes: non-infected control brains $n = 8$, infected control brains $n = 9$, non-infected Toll-1-RNAi brains $n = 7$, infected Toll-1-RNAi brains $n = 6$. **(H)** Illustration showing how adult-specific knock-down of *Toll-1* expression prevented cell death after infection. **(I, J)** Adult restricted knock-down of *Toll-1* in MyD88 cells improved **(I)** survival, **(J)** and climbing, at 7 days post-infection ($n = 82$–194 flies). **(I)** Two-way ANOVA survival: interaction: $p = 0.0458$; Genotype comparisons: $p = 0.0495$; Infected versus not-infected: $p < 0.0001$. **(J)** Two-way Anova climbing: interaction: $p = 0.0017$; Genotype comparisons: $p = 0.0177$; Infected versus not-infected: $p < 0.0001$. Dotted line in (A) indicates ROI for automatic cell counting with DeadEasy. Graphs in (C,E,G) show box-plots around the median, box with 50% of data points and whiskers with 25% of data points. Asterisks on graphs indicate multiple comparison correction tests: $*p < 0.05$, $**p < 0.01$, $****p < 0.0001$. Scale bars: (A, D, F) 100 μm. Genotypes: Control: *MyD88 GAL4 hisYFP/ + tubGAL80*ts/ + flies, whereby the GAL4 driver line was outcrossed to wild-type. Test flies: *MyD88 GAL4 hisYFP/UAS-Toll-1RNAi kk/100,078; tubGAL80*ts/+. Infected versus not-infected compares flies of the same genotypes. See S4 Table for further details. The data underlying C, E, G, I, and J can be found in S5 Table Source data.

cell loss requires Toll-1 signalling in MyD88+ cells. As MyD88 > hisYFP+ cells include glial cells of the BBB (S2B-E Fig), by inducing Toll-signalling dependent glial cell loss, *B. bassiana* could make holes in the BBB, making its way into the brain.

We also tested the effect of *Toll-1* RNAi knock-down in dopaminergic neurons (DANs). *Toll-1* is expressed in DANs, as there was colocalization of *Toll-1 > HistoneYFP* and anti-TH within multiple DAN clusters, including PAL, PPM3, PPL2, and PPL1 (S3A Fig), but in very few neurons within the PAM clusters (S3A Fig). Following *Toll-1* RNAi knock-down in MyD88 + cells, in non-infected flies, *Toll-1* knockdown did not alter DAN cell number (Fig 5F and 5G). By contrast, *Toll-1* RNAi knockdown prevented infection-induced neuronal loss within the TH + PPM3 and PPL1 DAN clusters (Figs 5F, 5G and S3B). Thus, DAN degeneration upon *B. bassiana* infection required Toll-1 (Fig 5H).

Finally, we tested whether acute adult-restricted *Toll-1* knock-down could rescue the behavioural impairments caused by *B. bassiana* infection. We found that *Toll-1* knock-down in MyD88 cells rescued fly survival after 7 days of infection compared to non-infected controls, albeit not significantly when compared to infected RNAi controls (Fig 5I). By contrast, *Toll-1* knock-down rescued the climbing impairment caused by *B. bassiana* infection, compared to infected genetic controls, but did not achieve the normal climbing performance of non-infected control flies (Fig 5J). Altogether, acute adult-restricted *Toll-1* RNAi knock-down improved fly survival and locomotion after infection. The rescues are consistent with the findings that *Toll-1* knock-down in adult flies prevents *B. bassiana*-induced neurodegeneration, whilst the incomplete nature of the rescues can be explained as immunity would be compromised in these flies.

Altogether, these data showed that *B. bassiana*-induced neurodegeneration depends on Toll-1.

### *B. bassiana* benefits from non-immune Wek-Sarm signalling to induce cell loss

Apoptotic Toll-1 signalling requires the adaptor Wek, which binds Sarm, which induces cell death [34]. As *B. bassiana*-induced the upregulation of *wek* and *sarm* in the fly brain (Fig 3G),

and B. *bassiana*-induced cell loss depended on Toll-1, we asked whether *wek* and *sarm* might also be required for infection-dependent cell loss.

First, we tested *wek*. In non-infected controls, adult-specific *wek* knock-down using line *UAS-wek* ^MH046534^-*RNAi* (shown to work effectively in [34,36]) had no effect on MyD88+ cells (Fig 6A and 6C). However, *wek-RNAi* knock-down rescued B. *bassiana*-induced MyD88-HisYFP + cell loss (Fig 6A and 6C). In non-infected control flies, *wek*-RNAi knockdown did not alter the number of Repo+ glial cells in the brain (Fig 6D and 6E). However, no difference was found in Repo+ glial cell number between infected and non-infected brains upon *wek*-RNAi knock-down (Fig 6D and 6E). These data mean that B. *bassiana* could not induce MyD88+ and Repo+ cell loss without Wek.

By contrast, *wek* knock-down in non-infected adult flies caused a significant decrease in the number of DANs of the PPL1, PPL2 and PPM3 clusters (Figs 6F, 6G and S4). This suggested that Wek is required to maintain DAN cell survival or promote neurogenesis or differentiation (e.g., TH expression) in the adult brain. These data are consistent with the known pleiotropic functions of *wek*, including in the adult brain [33,34,36]. Importantly, in *wek*-RNAi knock-down fruit-flies, B. *bassiana* infection failed to induce further neuronal loss (Figs 6F, 6G, S4A and S4B). These data revealed that in the absence of Wek, some DANs do not develop normally and only a few DANs remain, but these are not susceptible to further damage by B. *bassiana* infection. Finally, we tested whether *wek* RNAi knock-down could alter the behavioural impairments caused by exposure to B. *bassiana*. We found that *wek* knock-down in MyD88+ cells improved fly survival after seven days of infection, albeit not significantly (Fig 6I), and it did not rescue climbing (Fig 6J), which could be explained by DAN loss. Altogether, these data show that Wek is required for B. *bassiana*-induced loss of Myd88+, Repo+ and TH+ cells (Fig 6H).

Next we tested *sarm*. In the absence of infection, when we downregulated *sarm* expression with RNAi in adult MyD88 cells (using line *UAS-sarm-RNAi*^JF01681^, shown to work effectively in [34]), there was no effect (Fig 7A-C). However, RNAi knock-down of *sarm* rescued B. *bassiana*-induced MyD88 > hisYFP+ cell loss (Fig 7A-C). Similarly, *sarm* RNAi knock-down had no effect on glial cell number in the adult brain, but it rescued the Repo+ cell loss caused by B. *bassiana* infection (Fig 7D and 7E). And *sarm* RNAi knock-down did not affect PPM3 cell number in non-infected controls, but it prevented PPM3 DAN cell loss caused by B. *bassiana* infection (Fig 7F and 7G). Finally, *sarm* RNAi knock-down caused a mild albeit not significant increase in fly survival after seven days of infection (Fig 7I), and it improved climbing, albeit not significantly (Fig 7J). Altogether, these data showed that *sarm* is required for B. *bassiana*-induced loss of MyD88-HisYFP+, Repo+ and TH+ PPM3 cells (Fig 7H).

To conclude, these data show that signalling via Wek and Sarm is required for B. *bassiana*-induced cell loss in the brain. They suggest that the upregulation of Wek and Sarm upon fungal infection (see Fig 3G) could cause neurodegeneration in the host (Fig 7H).

## Increased *Toll-1*, *wek* and *sarm* levels can induce cell loss

We next asked whether over-expression of *activated Toll-1, wek* or *sarm* could be sufficient to induce cell loss in the absence of infection. Wek has pleiotropic functions downstream of Tolls: it binds Tolls and MyD88 to enable canonical signalling via NFκB/Dorsal/Dif downstream, which in the CNS promotes cell quiescence and cell survival; it can bind Sarm, to promote cell death; or function independently of both, to promote neurogenesis in the adult brain from quiescent MyD88 + progenitor cells [33,34,36]. Thus, the pleiotropic functions of *wek* could lead to compound phenotypes. To simplify this, we tested the effect of activated *Toll-1*^10b^, *wek* or *sarm* over-expression on PAM neurons, which are differentiated neurons and

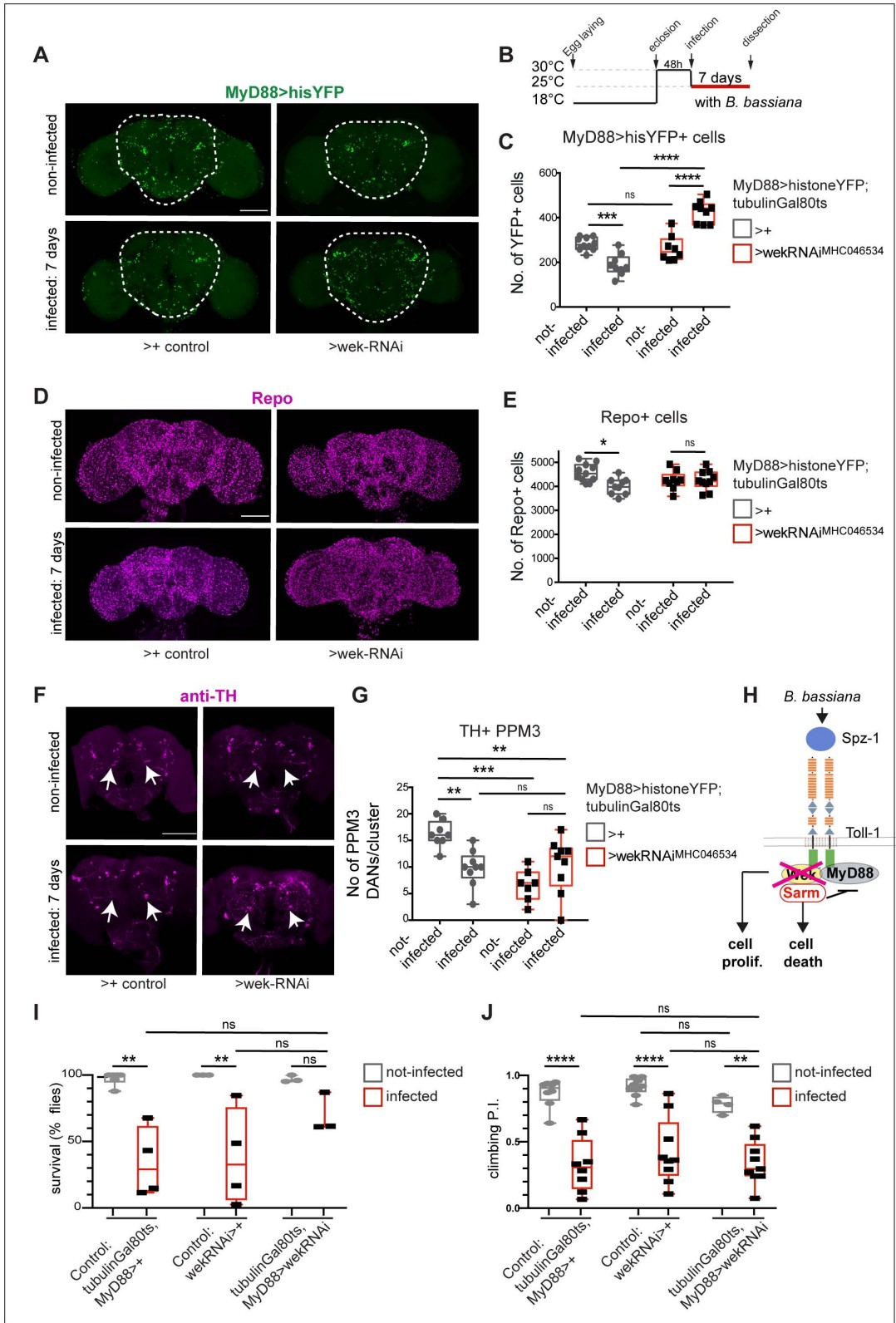

**Fig 6. *B. bassiana*-induced cell loss requires Wek, but Wek has pleiotropic functions. (A)** Loss of *MyD88 > HisYFP+* cells caused by seven-day exposure to *B. bassiana* was rescued with adult-specific *wek-RNAi*MHC046534 knockdown, and cell number increased further**. (B)** Diagram explaining the experimental regime. **(C)** Automatic quantification of data in

(A). Two-way ANOVA: Infected versus not-infected $p = 0.2327$; Genotypes: $p < 0.0001$; Interaction: $p < 0.0001$, followed by Tukey's multiple comparison correction test. Sample size: non-infected control brains $n = 14$, infected control brains $n = 15$, non-infected wek-RNAi brains $n = 8$, infected wek-RNAi brains $n = 9$. **(D, E)** Loss of Repo + glial cells caused by seven-day exposure to *B. bassiana* was rescued with adult-specific *wek-RNAi*MHC046534 knockdown, automatic quantification with DeadEasy in **(E).** Two-way ANOVA: Infected versus not-infected p = 0.0236; Genotypes: $p = 0.7775$; Interaction: $p = 0.0231$, and Tukey's multiple comparisons correction test. Sample sizes: non-infected control brains $n = 11$, infected control brains $n = 9$, non-infected *wek-RNAi* brains $n = 10$, infected *wek-RNAi* brains $n = 10$. **(F, G)** Adult-specific *wek-RNAi*JF01681 knock-down decreased the number of TH + PPM3 DANs (arrows in F), suggesting that *wek* may be required for their differentiation. In fact, *wek-RNAii*JF01681 knockdown did not rescue the cell loss caused by *B. bassiana* infection, but the infection did not reduce cell number further either. Two-way ANOVA: Infected versus not-infected: $p = 0.3098$; Genotypes: $p = 0.0016$; Interaction p = 0.0005, followed by Tukey's multiple comparisons correction test. Sample sizes: non-infected control brains $n = 8$, infected control brains $n = 9$, non-infected *wek-RNAi* brains $n = 7$, infected *wek-RNAi* brains $n = 9$. **(H)** Diagram of Toll signalling pathway, whereby Wek links Toll-1 to Sarm, enabling Toll signalling to cause cell death and cell loss via Sarm, and this is rescued in some cells with *wek RNAi* knock-down. Wek also has functions independently of Sarm and MyD88, promoting adult neurogenesis. **(I,J)** *wek-RNAi* knock-down in MyD88 cells had no effect on survival at 7 days post-infection **(I)**, nor in climbing **(J)** ($n = 73$–194). Two Way ANOVA survival: interaction: p = 0.2017; Genotype comparisons: p = 0.2437; Infected versus not-infected: $p < 0.0001$. **(J)** Two Way Anova climbing: interaction: $p = 0.7414$; Genotype comparisons: p = 0.2014; Infected versus not-infected: $p < 0.0001$. Dotted lines in (A) indicates ROI for automatic cell counting with DeadEasy software. Asterisks on graphs: * $p < 0.05$, ** $p < 0.01$, *** $p < 0.001$, **** $p < 0.0001$. Scale bars: (A, D, F) 100 μm. Genotypes: Control: *MyD88 GAL4 hisYFP/ + tubGAL80*ts/ + flies, whereby the GAL4 driver line was outcrossed to wild-type. Test flies: *MyD88 GAL4 hisYFP/UAS-wek-RNAi*TRIPHMC045534*; tubGAL80*ts/ + . Infected versus not-infected compares flies of the same genotypes. See S4 Table for further details. The data underlying panels C, E, G, I and J can be found in S5 Table Source data.

whose number can vary [79]. We used a DAN-specific GAL4 driver (*THGAL4; R58E02GAL4*), visualised PAMs with histone-YFP and counted them automatically. Toll-1 is normally expressed only in a fraction of PAM neurons (S3A Fig). Over-expressed activated *Toll-1*[10b] in DANs caused a mild and not significant decrease in PAM cell number (Fig 8A and 8B), suggesting that not many PAMs may express *wek, sarm* or both in the un-infected brain. To note, *B. bassiana* infection causes an up-regulation in *wek* and *sarm* expression (see Fig 3G). And over-expression of either *wek* (Fig 8A and 8B) or *sarm* (Fig 8C and 8D) was sufficient to decrease PAM neuron number in the absence of infection. Such targeted and partial DAN cell loss was not sufficient to impair climbing (S5 Fig). Altogether, these data show that increased Wek and Sarm levels can induce PAM cell loss. These data suggest that the upregulation of *wek* and *sarm* caused by *B. bassiana* infection leads to cell loss in the host brain.

## Discussion

We show that the fungus *B. bassiana* enters the brain, activates the innate immunity Toll-1 receptor, and up-regulates *wek* and *sarm* expression in the brain. Sarm is the inhibitor of Toll/TLR-dependent innate immunity, from flies to humans [26,37–40]. Through its ability to up-regulate Sarm, *B. bassiana* drives immune evasion, causing neurodegeneration in the host.

Our data show that *B. bassiana* infiltrated the brain, damaged the BBB, and activated Toll-1 signalling, upregulating the expression of antifungal peptides, *wek* and *sarm*, within infected, dissected, adult brains. The evidence that *B. bassiana* invaded the brain included *B. bassiana*-GFP + germlings or germinating spores within the brain. Entomopathogenic fungi have been demonstrated to enter the brains of their hosts, including *Drosophila* [3,80]. Spores in the brain and a compromised BBB were also found following *E. muscae* infection in *Drosophila* [3,80]. Our data are also consistent with RNAseq of *E. muscae*-infected brains showing the upregulation of innate immunity gene expression [3]. We do not know the sequence of events, but BBB breakdown could have started earlier than was detected. In fact, infection of *Drosophila melanogaster* with *E. muscae* induced BBB permeability, which became more severe over time. RNAseq revealed the up-regulation of Toll-dependent signalling at 24–48 h in infected brains, *E. muscae* fungal cells were detected within the brain at 48h post-exposure,

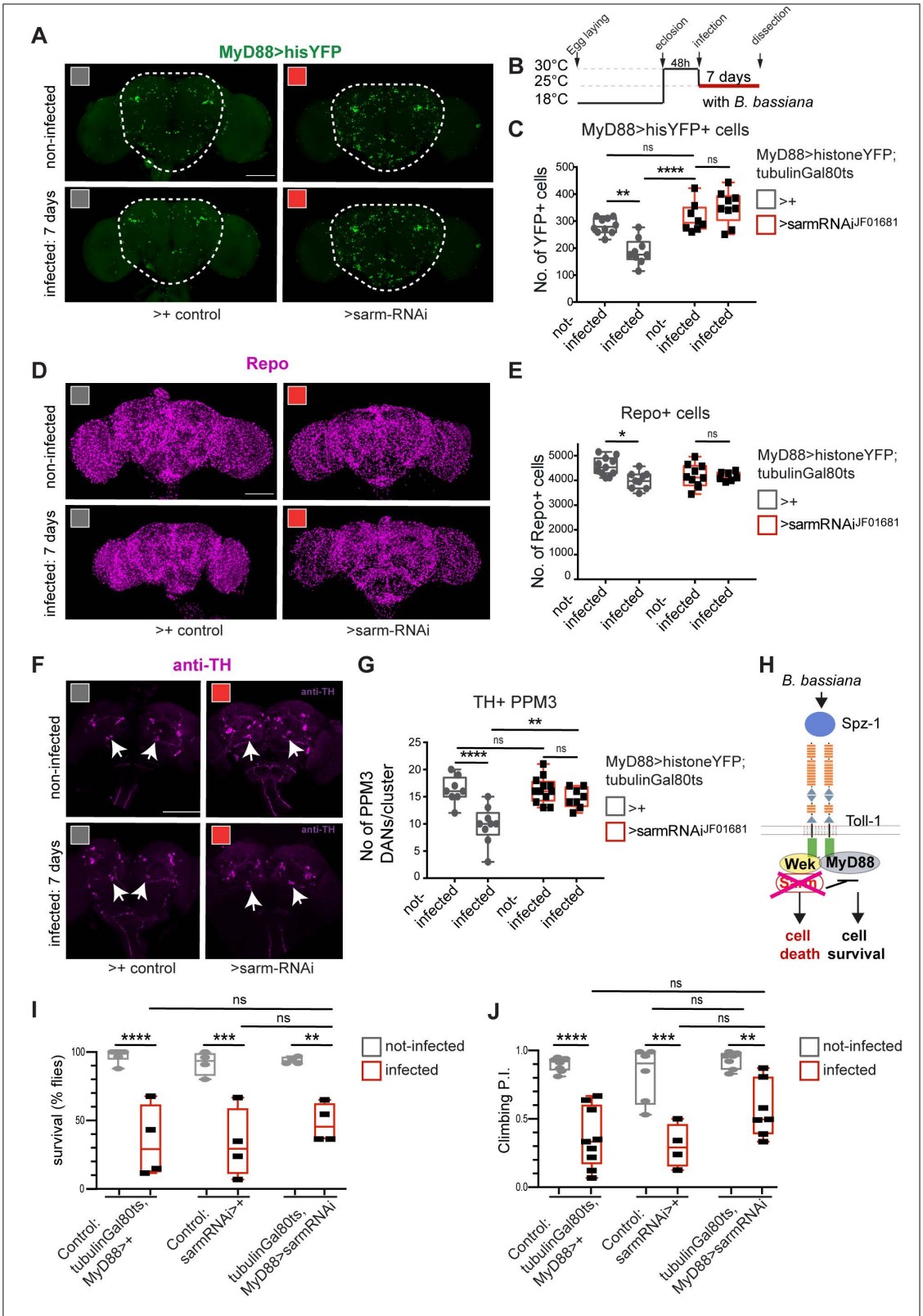

**Fig 7. *B. bassiana*-induced cell loss requires sarm.** **(A)** Seven-day exposure to the *B. bassiana* caused loss of *MyD88 > HisYFP+* cells, and this was rescued with adult-specific *sarm-RNAi*JF01681 knockdown, automatic quantification in **(C)**. **(B)** Diagram

explaining the experimental regime. **(A, C)** Two-way ANOVA: Infected versus not-infected p = 0.0077; Genotypes: $p < 0.0001$ Interaction: $p < 0.0001$, followed by Tukey's multiple comparison correction test. Sample sizes: non-infected control brains $n = 14$, infected control brains $n = 15$, non-infected sarm-RNAi brains $n = 8$, infected sarm-RNAi brains $n = 9$. **(D, E)** Seven-day exposure to *B. bassiana* caused loss of Repo + glial cells **(D)**, and this effect was rescued with adult-specific *sarm-RNAi*JF01681 knockdown, automatic quantification with DeadEasy in **I**. Two-way ANOVA: Infected versus not-infected p = 0.0205; Genotypes: $p = 0.3649$; Interaction: $p = 0.0295$, followed by Tukey's multiple comparisons correction test. Sample sizes: non-infected control brains $n = 11$, infected control brains $n = 9$, non-infected *sarm-RNAi* brains $n = 10$, infected *sarm-RNAi* brains $n = 7$. **(F, G)** Seven-day exposure to *B. bassiana* caused loss of TH + PPM3 DANs (arrows), which was rescued with adult-specific *sarm-RNAi*JF01681 knockdown, quantification in **(G)**. Two-way ANOVA: Infected versus not-infected $p < 0.0001$; Genotypes: $p = 0.0071$; Interaction: $p = 0.0071$, followed by Tukey's multiple comparisons correction test. Sample sizes: non-infected control brains $n = 8$, infected control brains $n = 9$, non-infected *sarm-RNAi* brains $n = 11$, infected *sarm-RNAi* brains $n = 8$. **(H)** Diagram of Toll signalling pathway, whereby Sarm leads to cell death and cell loss. *B. bassiana* infection caused cell-loss is rescued with *sarm* RNAi knock-down. **(I, J)** *sarm-RNAi* knock-down in MyD88 cells slightly improved survival **(I)** and **(J)** climbing albeit not significantly ($n = 76–194$ flies). Two-way ANOVA survival: interaction: $p = 0.5092$; Genotype comparisons: $p = 0.5591$; Infected versus not-infected: $p = 0.0001$. **(J)** Two Way Anova climbing: interaction: $p = 0.2992$; Genotype comparisons: $p = 0.0290$; Infected versus not-infected: $p < 0.0001$. Dotted line in (A) indicates ROI for automatic cell counting with DeadEasy software. Data in graphs are shown as box-plots around the median. Asterisks on graphs indicate multiple comparisons corrections: $*p < 0.05$, $**p < 0.01$, $****p < 0.0001$. Scale bars: (A, D, F) 100 µm. Genotypes: Controls: *MyD88 GAL4 hisYFP/ + tubGAL80*ts/+ flies, whereby the GAL4 driver line was outcrossed to wild-type. Test flies: *MyD88 GAL4 hisYFP/+ tubGAL80*ts/*UAS-sarm-RNAi*JF01681 Infected versus not-infected compares flies of the same genotypes. See S4 Table for further details. The data underlying panels C, E, G, I and J can be found in S5 Table Source data.

and BBB breakdown was most pronounced after 69h of exposure [3,80]. Thus, Toll signalling was detected first upon infection in the brain, followed by detection of fungal cells within the brain, and lastly detection of BBB breakdown [3,80]. This could reflect the different detection levels using the different techniques and that BBB breakdown started before it was detectable. In *Drosophila*, the BBB is formed by glia cells, and exposure to *B. bassiana* caused glial cell loss. We observed *B. bassiana-GFP* crossing the BBB at multiple points around the brain, and including the BBB separating the proboscis and oesaophagus from the brain. MyD88 is expressed in both neurons and glial cells [36], it contributes to the BBB, and infection caused MyD88+ cell loss. This means that *B. bassiana could* make its way into the brain through causing holes in the BBB by inducing glial cell death via Toll signalling. *B. bassiana* could enter the brain through feeding, as flies feed on *B. bassiana* and, and after feeding un-exposed flies with *B. bassiana-GFP*, *GFP*+ cells were found in the proboscis, at the point of entry and exit of the proboscis through the brain, and deep within the brain. In principle, fungal spores would not survive in the foregut and gut microbes can reduce the virulence of fungi against insects [39]. However, in mosquitos, *B. bassiana* invades the brain following ingestion and causes mosquito death only when ingested through the proboscis [81]. The proboscis traverses the brain, which could provide *B. bassiana* with an infection route that avoids entering the gut any deeper. In fact, it has been previously proposed that fungi may colonise the brain to create a fungal reservoir that avoids the hostile environment of the gut [3]. Fungi also invade the human brain [8,10,11,82,83]. The most common infection is by *Cryptococcus*, which causes meningoencephalitis, and *Cryptococcus* enters the human brain through the nose, by inhalation, and also degrades the BBB [8]. In summary, our data showed that *B. bassiana* invades the brain by crossing the BBB, which it can weaken by causing glial cell loss, and a likely route into the fly brain is through feeding.

Exposure to *B. bassiana* decreased fly longevity and impaired locomotion, correlating with widespread cell loss in the brain, including of dopaminergic neurons. Impaired survival and climbing upon infection could be due to multiple deficits in diseased flies, not only impaired brain function. Nevertheless, reduced longevity and impaired locomotion are commonly used indicators of neurodegeneration in fruit-flies [3,4]. *B. bassiana* infection also reduced the expression of TH, and dopamine is required for locomotion [64]. Our findings are consistent with reports that fungal volatiles induce neurodegeneration in the *Drosophila* brain [4,5].

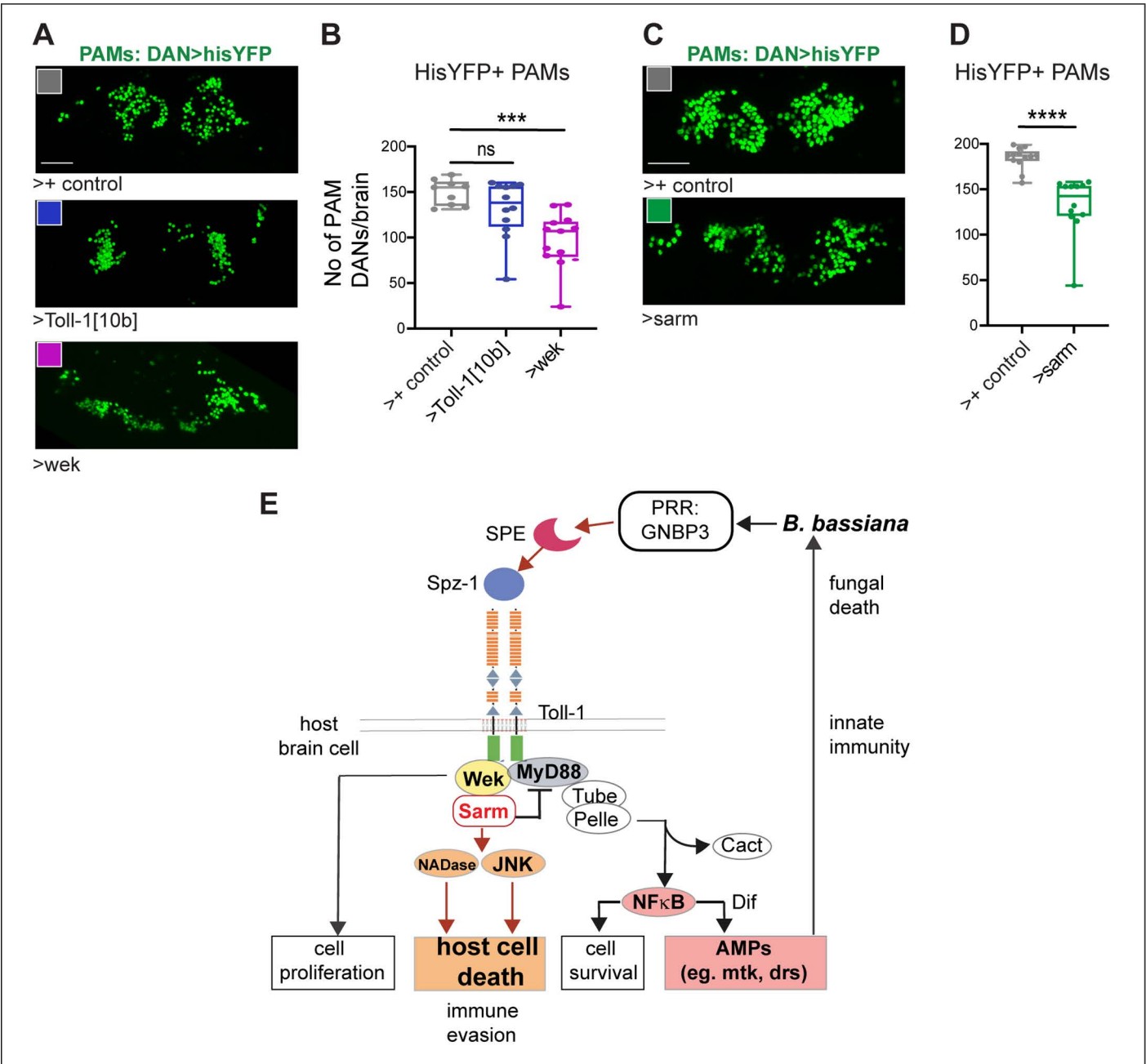

**Fig 8. Increased *Toll-1*, *wek* and *sarm* levels can induce cell loss in the absence of infection. (A,B)** In the absence of infection, over-expression of activated *Toll-1*10b in DANs (with *THGAL4; R58E02GAL4*) caused a rather mild and not significant decrease in PAMs. By contrast, over-expression of *wek* was sufficient to induce cell loss in 7-day-old flies. Sample sizes: control $n = 9$ brains, *UAS-Toll-1*10b $n = 12$; *UAS-wek-HA* $n = 13$. One-way ANOVA $p = 0.0003$, multiple comparisons corrections Dunnett test to a fixed control. **(C,D)** Over-expression of *sarm* was sufficient to induce PAM cell loss in the absence of infection in 2-day-old flies. Sample sizes: control $n = 13$ brains, *UAS-sarm* $n = 12$. Student $t$ test. Asterisks on graphs indicate multiple comparisons corrections: ****$p < 0.0001$. Scale bars: (A, C) 30μm. Flies were kept constantly at 25°C. **(E)** Diagram illustrating the multiple signalling pathways downstream of Toll-1 that can be activated by *B. bassiana*. Signalling via MyD88 and Dif/NFκB results in fungal elimination, whereas signalling via Wek and Sarm inhibits MyD88-dependent innate immunity and drives host cell death instead. Genotypes: Controls: *THGAL4; R58E02GAL4/+* flies, whereby the GAL4 driver was outcroseed to wild-type. Test flies: *THGAL4/UAS-Toll-1*10b; *R58E02GAL4/+*; *THGAL4; R58E02GAL4/UAS-wek-HA*; *THGAL4; R58E02GAL4/UAS-dsarm*. See S4 Table for further details. The data underlying panels B and D can be found in S5 Table source data.

Evidence of neurodegeneration were decreased longevity, impaired climbing, decreased dopamine levels and dopaminergic neuron loss [4,5]. These same phenotypes are also measures of neurodegeneration in *Drosophila* models of Parkinson's disease [63,64,84]. And fungal infections in the human brain correlate with parkinsonism [10,11,85]. Altogether, exposure to *B. bassiana*-induced a phenotypic signature characteristic of neurodegeneration.

Our data showed that *B. bassiana*-induced cell loss requires Toll signalling via Wek and Sarm. Wek enables Toll signalling via Sarm, as Toll-1 cannot directly bind Sarm, and instead it binds Wek, which binds Sarm [33,34]. *B. bassiana* infection caused an increase in *wek* and *sarm* expression within dissected brains, clear of any surrounding cells. Similarly, *E. muscae* infection also induced the upregulation of *wek* [3]. We showed that knocking-down *Toll-1, wek* or *sarm* rescued the cell loss caused by exposure to *B. bassiana*. A caveat is that testing knock-down of an unrelated gene could have controlled for potential non-specific effects of RNAi. Nevertheless, knock-down of *Toll-1, wek* and *sarm* gave consistent results with multiple cell markers, demonstrating that neurodegeneration required the activation of this pathway. It had been shown before that Toll-1, Wek and Sarm can drive apoptosis [34]. This could explain why knocking them down not only rescued the cell loss caused by the infection, but even increased cell number further, as this could rescue cells that would normally die. Importantly, we showed that although *Toll-1* expression varies among different DAN clusters, *Toll-1* knock-down rescued *B. bassiana*-induced loss of cells expressing *Toll-1*. The variable effects in distinct DANs also suggest that different DAN clusters may express different combinations of Tolls, downstream adaptors and effectors driving distinct outcomes. In fact, different Tolls can have distinct effects on survival or death [34]. It has been proposed that Toll-1 and -7 maintain DAN cell survival via regulating autophagy, through MyD88 [86], meaning that PAMs normally express *MyD88*. By contrast, *Toll-1^{10b}* could not induce severe PAM cell loss, which can be explained as not all PAMs normally express both *wek* and *sarm* [79]. Absence of Wek in PAMs would prevent Toll-1 from driving apoptosis, even in the presence of Sarm [34]. Instead, over-expression of *wek* or *sarm* was sufficient to induce PAM cell loss. Our data also suggest that *wek* is required for cell survival, neurogenesis or differentiation, consistent with the fact that *Toll-2* promotes adult neurogenesis via Wek [36]. Data support the notion that Wek has pleiotropic functions that depend on cellular context [33,34].

Interestingly, adult-restricted knock-down of *Toll-1* in MyD88+ cells improved fly survival after seven days of infection (albeit not significantly compared to infected genetic controls). By contrast, *Toll-1* mutant flies pricked with an infected needle survived only for two days [16]. This difference could be due to pricking causing a faster, more severe and systemic infection than exposure, and *Toll-1* mutants lacking both Toll-1 and immunity throughout development. Instead, acute Gal4-dependent RNAi knock-down in MyD88 cells only and restricted to the adult causes hypomorphic phenotypes. Remarkably, acute adult restricted *Toll-1RNAi* knock-down improved the climbing impairment caused by *B. bassiana* infection, when compared to infected genetic controls, but it did not rescue performance to the normal levels of non-infected controls. The incomplete nature of such rescues can be explained as acute *Toll-1* knock-down would weaken immunity throughout the body, rendering flies diseased compared to controls. Most remarkably, the improvements in longevity and locomotion upon *Toll-1* knock-down in infected flies, support the notion that *B. bassiana*-induced neurodegeneration via Toll-1.

The data also suggest that signalling by Toll-1/Wek/Sarm may not be enough to induce widespread neurodegeneration after infection. Perhaps other Tolls might also be involved. Alternatively, anti-microbial peptides induced by Toll-1 via MyD88/NFκB/Dif signalling could also drive neurodegeneration [87]. This would explain why knock-down of *Toll-1* rescued the survival and climbing impairment in infected flies (compared to some controls)

despite impaired immunity, whereas the effects of knocking down *sarm* were less robust. In fact, anti-microbial peptides regulated by NFκB Rel downstream of the Imd pathway can induce neurodegeneration [88]. In *C. elegans*, TIR-1/Sarm promotes the expression of anti-microbial peptides in the epidermis, which cause degeneration of neuronal dendrites [89]. Furthermore, fungal volatiles alone are sufficient to drive neurodegeneration [4,5], although whether the required high concentrations are encountered in nature remains to be discovered. Most likely, *B. bassiana*'s ability to induce neurodegeneration in the host may involve a combination of such mechanisms.

Activation of Sarm signalling by *B. bassiana* is likely one of a range of adaptive interactions with its hosts. In mosquitos, *B. bassiana* can reduce the host immune response by exporting a microRNA that downregulates the levels of Spz4, a ligand of Toll, in the fat body [90]. Similarly, in mammals, Toll-Like Receptor-4 (TLR-4) activates the immune response against *C. albicans,* while *C. albicans* activates TLR-2, which leads to production of anti-inflammatory chemokines that help *C. albicans* to evade the host immune system [91–93]. Our findings are consistent with the notion that *B. bassiana* drives immune evasion by manipulating the Toll pathway. We have shown that *B. bassiana* can induce Sarm signalling harming the host. What controls the up-regulation of *wek* and *sarm* after infection is unknown, but it could be Toll signalling itself, activated by *B. bassiana*. In fact, Tolls can up-regulate the expression of the downstream targes *dorsal* and *dif*, and their inhibitor *cactus* [35]. Importantly, the functions of Sarm in inhibiting canonical Toll/TLR signalling and inducing neuronal apoptosis and axon destruction are highly evolutionarily conserved [26,37,38,47]. In flies and mammals, Sarm induces apoptosis via JNK signalling and in *C. elegans*, via MAPK/p38 signalling [34,39–41,44–46,48,94]. Moreover, Sarm has a TIR domain that has catalytic NADase activity, which drives neurite destruction as well as cell death [44,46,95]. Most intriguingly, similarly to Sarm, some prokaryotic proteins bear a TIR domain with NADase activity that functions in immune evasion [47].

To conclude, a novel tactic in the evolutionary arms race between host and fungus takes place in the brain. *B. bassiana* is detected in the fly brain, this activates Toll-1-dependent innate immunity, but *B. bassiana* benefits from the concomitant activation of the immune-evasion Sarm pathway driving neurodegeneration in the host. Importantly, human neurodegenerative and psychiatric diseases have been linked to fungal infections and neuro-inflammation [8–14]. Sarm has also been linked to neurodegenerative diseases [94,96–98]. For example, constitutively active Sarm1 variants are enriched in ALS patients [96]. It will be important to find out whether a similar activation of Sarm downstream of TLRs in response to fungal infections is responsible for inducing psychiatric and neurodegenerative diseases in humans.

## Materials and methods

### *Drosophila* genetics

Please see S1 Table for the list of the stocks used. Conditional over-expression and knock-down in adult flies were carried out using ubiquitously expressed *tub-GAL80$^{ts}$*. *GAL80$^{ts}$* is a temperature-sensitive GAL4 repressor that prevents GAL4 expression at 18°C and enables it at 30°C. The temperature regimes we used are indicated in the figures and were set to enable GAL4 adult-onset and *Beauveria bassiana* growth together. Controls used were standard F1 outcrosses of GAL4 or UAS lines to wild-type (Oregon R in our work), except for Fig 1, where controls were the F1 progeny of wild-type Oregon crossed to wild-type Canton-S. Throughout the paper, the symbol ">" stands for F1 flies bearing both a GAL4 driver and a UAS responder. The symbol ">+" stands for F1 from GAL4 driver crossed to wild-type. Details of genotypes

and sample sizes are provided in S4 Table. Both females and males were used for experiments in Figs 1–4 (exactly the same number of each); only females were used for all other data.

### *Beauveria bassiana* culture, spore collection and infection chambers

*Beauveria bassiana* (80.2 strain) was a gift from Jean-Marc Reichhart and Jean-Luc Imler, (IBMC/University of Strasbourg); GFP transgenic *B. bassiana-GFP* (EABb 04/01-Tip GFP5 strain) was contributed by Prof. Enrique Quesada-Moraga (University of Cordoba).

**Spores.** To isolate *Beauveria bassiana* spores, 10 ml of distilled water was poured on the *B. bassiana* culture in the petri plate and culture was scraped with a spreader. The solution was centrifuged at 4,000 rpm for 15 mins, the supernatant was removed, the pellet was dissolved in 2–5 ml of water and spun again for 1–2 mins. The pellet was dissolved in 1 ml distilled water and labelled as the principal solution. The concentration of the principal solution was calculated by counting spores using a haemocytometer. 10 μl of principal solution was diluted in 90 μl of water and marked as dilution X10 and spores were counted again by using the haemocytometer. This step was repeated until a spore concentration of $3.7 \times 10^9$ spores/ml was achieved.

**Infection chamber.** To enable natural infection whilst avoiding damage to the body that would induce an injury response, a natural infection chamber was devised. An infection chamber consists of a glass bottle with *B. bassiana* growing on sabouraud dextrose agar (SDA) medium at the bottom, and a cut 50 ml falcon tube bearing standard fly-food stuck with double-sided sticky tape to the bottle side wall. This separate compartment for fly-food was affixed to the bottle's side wall, to prevent fungal growth from limiting food and hydration supply (Fig 1A). To prepare SDA media, 16.25 g of 4% SDA was added to 250 ml of distilled water and autoclaved. Next, 10 ml SDA medium was transferred into autoclaved glass bottles and once it had set, 1 ml of $3.7 \times 10^9$ spores/ml *B. bassiana* solution (see above) was poured into each bottle onto the medium. After plugging the bottles, they were transferred to 25°C incubator for 7–10 days. Next, the falcon tube containing fly-food was inserted into the bottle, and then the flies. In this way, flies can freely access fly-food devoid of *B. bassiana*, for feeding and hydration, whilst being naturally exposed to spores. The 10-day-old fungal culture included both hyphae and spores. Two-day-old flies were inserted into the bottle and shaken, enabling flies to get directly exposed to spores. Flies in the bottles were then kept at 25°C for several days depending on the experiment (e.g., 1, 3, 7), after which only active flies were selected for further tests. For the longevity test (Fig 1B), fungal cultures were replaced every 10–15 days.

### *Drosophila* behaviour tests

**Longevity** was measured as described in [99], using F1 wild-type Oregon/CantonS flies raised on standard cornmeal food. They were placed in an infection chamber, the fly food was replaced every four days in both control and experimental setups, and the number of dead flies and censor events were manually recorded after every transfer. Both infected flies and non-infected controls were kept at 25°C. Dead flies were scored as 1 and flies that got stuck to the food or escaped were scored as 0. Data were analysed using GraphPad prism and Log rank (Mantel-Cox test) to analyse the data and generate a Kaplan-Meier survival curve. The experiment was conducted with $n = 104$ flies for each condition, from two biological replicates of $n = 58$ and $n = 46$ flies each.

**Climbing assay.** Startled-induced negative geotaxis (SING) or climbing assay was carried out as described in [100]. The test was carried out in a humidity and temperature-controlled lab (25°C). Following exposure to *B. bassiana* in infection chambers, only active flies were

selected for climbing assays. Flies that were immobile, dying or stuck to the fungal culture were not used (see S1 Video). To prepare flies for climbing experiments, infected flies were first transferred from the infection chamber to standard vials containing fly-food, and they were instantly flipped 2–4 times, with 20–30 sec between flips, to remove excess spores from their cuticles. Then they were transferred to climbing assay vials, and after 30 mins of habituation, the climbing experiments were performed. The vials with flies inside were tapped, and the flies were filmed for 10 seconds, followed by a 30-second rest period, and the number of flies climbing above the 2 cm mark were counted (see S1 Video). This process was repeated 15 times for each cohort. At least three cohorts of 7–10 flies each were used per genotype, and the entire process was repeated a second time using new flies. Performance index (PI) = $1/2[(n_{tot} + n_{top} - n_{bot})/n_{tot}]$, where $n_{tot}$, $n_{top}$, and $n_{bot}$ are the total number of flies, the number of flies above 2 cm, and the number of flies below 2 cm, respectively.

**Proboscis extension response (PER) assay.** The PER assay was adapted from [101]. Flies were starved in a 25°C incubator for 24 hours in vials containing agar before performing the PER assay. After 24 hours, the flies were transferred to a 20 μl pipette tip using a fly aspirator. Using a sharp razor blade, the tip was cut, and the fly was then carefully placed in position with the help of a wick made of Kimtech wipe. The 20 μl tip with the fly head emerging was fixed on a flat surface and a camera was positioned around it. To verify that the immobilized flies in the 20 μl tip were fit for the PER assay, they were first given water and 100 mM sucrose solution using tissue paper wicks and the response was recorded for 60 seconds. The flies had been hydrated but starved before the experiment, so any flies that responded to water or did not respond to the 100 mM sucrose solution were discarded.

To activate neurons, we used *TrpA1* over-expression. TrpA1 is a cation channel normally closed at lower temperatures, and it opens at 30°C, letting Na$^+$ and Ca$^{2+}$ into the neuron, triggering an action potential. In transgenic flies, this makes it possible to activate any neurons of choice, under the control of GAL4, by moving flies from 18 to 30°C. The PER assay was conducted inside an environmental chamber at 30°C. After the flies had been inside the chamber for 2–3 min, they were recorded for 60 seconds. The experiment was conducted using 10 flies per genotype, and the process was repeated three times using new biological replicates of the crosses.

**Feeding assay and measurement of ingested blue dye.** The feeding assay was adapted from [102]. To prepare a fly cage, an empty food vial was used, and was cut at 4 mm from its base using a heated knife. The base was then sterilized with 70% ethanol. Solutions mixed with 4% brilliant blue dye were transferred to the base and placed in an egg-laying chamber. For the feeding assay, four solutions were prepared, with each egg-laying chamber containing only one solution: Negative control 1: distilled H$_2$O; Negative control 2: distilled H$_2$O (950ul) + 50ul 4% brilliant blue dye (95%/5% v/v); Positive control: 100mM Sucrose solution (950ul) + 50ul 4% brilliant blue dye (95%/5% v/v); Test: $7.0025 \times 10^7$ spores/ml (950ul) + 50ul 4% brilliant blue dye (95%/5% v/v). Equal number of wildtype Oregon male and female flies were used ($n = 10$ males and 10 females) and 4 biological replicates were performed, flies were anesthetized on ice and transferred to the 4-feeding assay setups containing the above solutions. The set-ups were then placed at 25°C incubator for 24 hrs. Next, the flies were collected by anesthetising the flies with CO$_2$ and transferred to the 1 ml Eppendorf tube. The Eppendorf-containing flies were dipped in liquid nitrogen for 5 secs, placed in the 50 ml Falcon tube, and dropped from a height to separate the flies' heads from their abdomens. The abdomen of the flies was collected using forceps and transferred to an Eppendorf containing 50ul of distilled water. Using a pestle abdomen was homogenised in the Eppendorf and 950ul of dH$_2$O was added to it. The Eppendorf was then vortexed and centrifuged for 5 mins at 10,000 rpm. The supernatant was transferred into the new Eppendorf and kept on ice until

analysis. The homogenised solution was transferred to 1 ml cuvette, and the absorbance was recorded using spectrophotometer, at 630 and 750 nm. The corrected absorbance was calculated by subtracting the mean of 3 readings at 750 nm (background) from the mean of 3 readings at 630 nm (peak of blue dye).

**Feeding flies with *B. bassiana*-GFP.**  The feeding chamber was adapted from [102]. To prepare a fly-feeding cage, an empty food vial was cut 4 mm from its base using a heated knife. The base was then sterilized with 70% ethanol. A water-spore mixture with a concentration of $3.7 \times 10^9$ spores/ml was poured into the base and placed in an egg-laying chamber. The setup was then placed at 25°C incubator for 24 hours. Afterwards, flies were collected, anesthetised with $CO_2$ and immersed in 70% ethanol for 1 min. The flies were subsequently washed with 0.5% PBT for 2 min, followed by a final wash with PBS for 1 min. The brains of adult flies were dissected in PBS and transferred to 4% paraformaldehyde in PBS for 20 min. The brains were then washed 5 times with 0.5% PBT for 20 min each. Finally, the brains were rinsed with PBS for 20 min and mounted on slides using Vectashield for confocal microscopy.

## Staining of *B. bassiana* spores

*B. bassiana* cells were visualised after 3 days of exposure, using: (1) FM4–64, a red fluorescent dye (excitation/emission maxima ∼515/640 nm) used to stain yeast vacuolar membranes [103]. FM4–64 dye has been reported to stain *B. bassiana* spores in-vivo in the haemolymph of tobacco budworm [104], suggesting its potential use in staining spores in the adult Drosophila brain. (2) Calcofluor white (CLW, which stains cell walls of algae, plants and fungi), a fluorescent dye that binds to 1–3 beta and 1–4 beta polysaccharides of chitin and cellulose present in fungal cell walls [105]. (3) Transgenic *B. bassiana*-GFP, used after 1 day of exposure, as flies died earlier with this strain. Adult *Drosophila* brains were dissected and fixed in 4% paraformaldehyde (PFA) in Phosphate buffer saline (PBS) at room temperature for 20 min, taking great care to remove fat body and all other surrounding cells and tissues. Following washes in 0.5% Triton in PBS, brains were incubated in FM4–64 dye at 1:200 dilution, or calcofluor white (50 mg/ml) at 1:1,000 dilution for 1h at room temperature. The brains were washed 20 mins in 0.5% PBT five times plus a final wash in PBS and mounted in DAPI containing Vectashield, prior to imaging.

## Injecting flies with dextran red fluorescent dye

This protocol was adapted from [67]. A glass-pulled capillary needle was inserted into a glass pasteur pipette and sealed with parafilm at the point of insertion, and needle was loaded with 0.1 μl of 25 mg/μl fluorescent Dextran red dye. The needle was carefully inserted into the thorax beneath the wing of anaesthetised flies. The flies were then transferred to a vial containing fly food and placed in a 25°C incubator for 24 hrs to recover. After the 24 hours recovery period, the flies were fixed onto a glass slide using glue, and their retinas were imaged using Leica SP8 confocal fluorescent microscope, $n = 10$ flies.

## Quantitative reverse transcription (qRT)-PCR

qRT-PCR was carried out from 20 wild-type (Oregon) adult brains per sample, dissected from wild type Oregon non-infected and infected flies. The dissected adult brains were thoroughly cleaned removing all fat body, other surrounding cells and tissues, and then immediately transferred into the Eppendorf containing TRI reagent (Ambion #AM9,738). RNA extraction was carried out according to the TRI reagent protocol using isopropanol. Extracted RNA was treated with DNase to get rid of genomic DNA contamination using DNA-free kit (Ambion #AM1906). 200 ng of RNA was reverse transcribed into cDNA by using random primer by

following the manuscript of GoScript™ Reverse Transcription System (Promega #A50001) (S2 Table). A PCR was performed to check the contamination in the RNA and Reverse Transcribed (RT) cDNA (S2 Table). This is followed by a qRT-PCR. All qRT-PCR experiments were done in triplicate (i.e., 3 well condition). A DNA-binding dye SYBR was used to observe the dynamics of the PCR (SensiFast™ SYBR) and used ABI Prism 7,000SDS machine for the qPCR (S2 Table). Following qRT-PCR, quantification was performed using the CT value generated by the qPCR machine, as described in [106]. For the internal control, GAPDH (housekeeping\reference gene) was used. The expression of the target gene (ΔCT) was normalised in infected and non-infected flies by subtracting the CT value of the GAPDH (reference gene) from the CT value of the target gene. ΔΔCT was calculated by subtracting the ΔCT (Target gene) of the infected flies from the ΔCT (Target gene) of the non-infected flies. The relative gene expression of the target gene was then calculated in infected and non-infected flies and expressed as fold change ($2^{-\Delta\Delta CT}$). The statistical analysis was conducted on the ΔCT values. Three biological replicates were used each biological replicate consisted of 20 flies. ΔCT = CT (a target gene) – CT (a reference gene\GAPDH); ΔΔCT = ΔCT (non-infected sample) – ΔCT (infected sample); Fold change = $2^{-\Delta\Delta CT}$

## Generation of Toll-Gal4 flies

Toll-1-GAL4 transgenic flies were generated using CRISPR/Cas9 enhanced homologous recombination. The 5′ homology arm was PCRed using primers 5'Fwd: ATGCGACCG-GTAAAATCTCGTATTATGCAGCACTCGA and 5'Rev: GGAACTGAGCGGCCGCT-GCAAATGGAGAAATTGAAAGGAAT; the 3′ homology arm was PCRed using primers 3'Fwd: GATGGCGCGCCGTGAACCCATTTGGACAACA and 3′ Rev: CGTACTAGTG-CAGTTCAGCTCTCAGCCGT. The donor vector was pT-GEM-T2A-Gal4, and homology arms were cloned into AgeI and NotI (5′ homology arm) and AscI and SpeI (3′ homology arm) enzyme sites (as in [107]). The guide RNA targeted the start of the coding sequence: sense oligo: GTCGCCCATTTGGACAACATGAGTCGA; antisense oligo: AAACTCGACT-CATGTTGTCCAAATGGG. The gRNA was cloned into the BbsI site of vector pU6.3, as described in [108]. Transgenesis was carried out by the BestGene. 3xP3-DsRed was subsequently removed with CRE-recombinase using conventional genetics. Primers are given in S2 Table.

## Immunostainings

To prepare flies for immunostaining, **i**nfected flies were first transferred to vials from infection chamber and flipped twice to remove excess spores. The flies were then transferred to 70% ethanol and washed in 1XPBS for 10 min. Adult Brains were dissected in 20 min and fixed in 650ul fixative (650ul 4% paraformaldehyde (PFA) in 1× Phosphate buffer saline (PBS) at room temperature for 20 min in a 1.5 ml Eppendorf. The dissected adult brains were thoroughly cleaned removing all fat body, other surrounding cells and tissues. Subsequently, the brains were washed for 20 mins in 750ul of 0.5% PBT (47.5 ml 1xPBS + 2.5 ml 10% Triton) 5 times. Following this, the brains were incubated in anti-TH (at 1:20) or anti-Repo (at 1:250) primary antibodies for 2 days. Following this, brains were washed for 20 min 5 times in 0.5% PBT, then incubated in fluorescently tagged secondary antibodies (anti-Mouse for Repo and anti-Rabbit for TH) overnight, as and if appropriate. After this, brains were washed again in PBT as above. Finally, the brains were mounted onto slides with either Vectashiled or DAPI containing Vectashield depending on the experiments and kept at 4°C overnight ready to be scanned. Antibodies used, their source and their working dilutions are given in S3 Table. We used both females and males for experiments in Figs 1–4, but we used exactly the same number of each

gender for the experiments. We did not observe any different effects between the sexes. Thus, for all other experiments, we used female flies only, because they are bigger, so their brains are also bigger and easier to dissect.

## Microscopy and imaging

*B. bassiana* spores in the adult brain stained with FM4–64 dye, Calcofluor white stain, and transgenic *B. bassiana*-GFP spores were imaged with Leica SP8 confocal microscope with a 20X oil immersion lens, at a resolution of 1024 × 1024 pixels, zoom 1 and scanning speed of 400 Hz, and 0.96 μm Z step. High magnification images of *B. bassiana* spores in adult brains were obtained using a 63X oil immersion lens at 1024 × 1024, zoom 2.0, speed of 400 Hz, line average of 4 and Z step 0.96 μm.

Diffusion of Dextran Red dye in the retina of adult flies was imaged using Leica SP8 confocal microscope with a 10× lens, at a resolution of 1024 × 1024 pixels. The zoom was set to 0.8 and scanning speed was 400 Hz.

Toll-1 > FlyBow1.1, Sarm>FB1.1 proboscis were imaged using Zeiss LSM Airyscan confocal microscope, with 10X objective, 1024 × 1024 resolution, zoom 1, speed 5 fps, line average 4 and Z step 1 μm. The resulting images were produced using ImageJ software.

Brains for cell counting were imaged as follows: Anti-Repo and SarmNP0257 > hisYFP were imaged using a Zeiss LSM 710 confocal microscope with a 25x oil immersion lens, at a resolution of 1024 × 1024 pixels, zoom 0.6 or 1, speed of 7 frames per second (fps), line average of 1 and 0.96 μm step. MyD88 > hisYFP samples were imaged using a Leica SP8 confocal microscope with a 20X oil immersion lens, at 1024 × 1024, zoom 1, speed 400 Hz and Z step 0.96 μm.

Anti-TH+ stained Toll- > HisYFP adult brains were imaged with the Leica SP8 confocal microscope with a 20X oil immersion lens, at 1024 × 1024 pixels, zoom 1.4, speed 400 Hz and Z step 0.96 μm. For anti-TH in wild-type brains, the Zeiss LSM 710 confocal microscope was used, with a 25x oil immersion lens, 1024 × 1024 at 8 fps, line average of 1 and Z step 0.96 μm. For MyD88 > hisYFP anti-TH samples, adult brains were scanned with the Leica SP8 confocal microscope with a 20X oil immersion lens, at 1024x1024, zoom 1.4, speed 400 Hz and Z step 0.96 μm.

## Quantification of cell number using DeadEasy

Cells were counted automatically using DeadEasy software, developed as ImageJ plug-ins, as reported in [36]. Repo+ cells in the adult brain were counted automatically with the *DeadEasy Glia Adult*; Sarm>HisYFP+ and MyD88 > HisYFP+ cells were quantified with *DeadEasy adult central brain*; and THGAL4 R58E02GAL4 > HisYFP+ cells were counted with *DeadEasy adult central brain*. Dopaminergic neurons labelled with anti-TH were counted manually, assisted by the Fiji cell counter. DeadEasy plugins published and previously used in [36,79] are publically available through UBIRA https://edata.bham.ac.uk/; https://doi.org/10.25500/edata.bham.00001213.

## Statistical analysis

Data were collected using *Excel* (Microsoft) and analysed using GraphPad Prism. Longevity assay data were analysed using Log Rank (Mantel-Cox test). Categorical proboscis extension response data were analysed using Chi-square test. Cell number counting data were numerical and continuous. Datasets with four or more groups were first tested for normality. For sample groups with two sample types, unpaired Student $t$ -tests were conducted, and Mann-Whitney U tests if not normally distributed. For comparisons involving infection versus, not-infected,

and two genotypes, two-way ANOVA followed by Turkey's multiple comparison tests was used, using 95% confidence.

## Supporting information

**S1 Fig. Not-exposed FM4–64 and *B. bassiana-GFP* controls. (A)** Wild-type brains not exposed to fungi, treated and stained with FM4–64, at the same time and in the same way as flies exposed to *B. bassiana* in Fig 2A. Tracheae are often visible in confocal stainings of the brain producing no specific signal. **(B)** Wild-type brains not exposed to fungi, stained with anti-GFP, at the same time and in the same way as flies exposed to *B. bassiana-GFP* shown in Fig 2C-E. Tracheae are often visible in confocal stainings of the brain producing no specific signal and anti-GFP can give a background signal.
(TIF)

**S2 Fig. Expression of *Toll-1, MyD88* and sarm in the brain. (A)** *Toll-1 > mCD8-GFP, MyD88^{NP6,394} > myrGFP* and *sarm^{NP7,460} > FlyBow* reveal that these genes are expressed widely throughout the adult brain. **(B)** *MyD88 > DenMark* reveals the BBB surrounding the brain, detail in **(B')**. **(C,C')** Colocalisation of nuclear markers YFP and the pan-glial nuclear marker anti-Repo (arrows) along the BBB, in *MyD88 > hisYFP* flies. The BBB is Drosophila if is formed of glial cells, and MyD88+ glia line up the outer edge of the brain (arrows), detail in **(C')**. **(D)** Detail from brains in **(A,B)** showing that *Toll-1 > mCD8-GFP, MyD88^{NP6,394} > DenMark* and *sarm^{NP7,460} > FlyBow* are localised at the entry point (anterior) of the proboscis into the brain and exit (posterior brain) of esophagous and neuropiles from the brain (arrows). **(F)** Colocalisation of YFP and Repo in glial cells of the BBB between proboscis and esophagus entry/exit through the brain (arrows) in *MyD88 > hisYFP* flies.
(TIF)

**S3 Fig. Toll-1 is expressed in PAMs and is required in PPL1. (A)** Co-localisation of *Toll-1 > histone-YFP* with anti-TH in PPL1, PPL2, PPM3, PAL and a subset of PAM DANs, in adult fly brains. **(B)** Adult-restricted *Toll-1-RNAi* knock-down in MyD88 + cells does not affect PPL1 neurons, but it rescues the loss of PPL1s caused by *B. bassiana* infection. Two-way ANOVA: Infected versus not-infected $p = 0.9586$; Genotypes: $p = 0.3628$; Interaction: $p = 0.0024$, followed by Tukey's multiple comparisons correction test. Sample size *Toll-1 > histone-YFP* fly brains n = 5, UAS-histone-YFP/+, $n = 5$, non-infected control brains $n = 8$, infected control brains $n = 9$, non-infected *Toll-1* ^{KK/100,078} *RNAi* brains $n = 7$, infected *Toll-1* ^{KK/100,078} *RNAi* brains $n = 6$. Graphs show box plots around the median. Asterisks on graphs: *$p < 0.05$, **$p < 0.01$, ***$p < 0.001$, ****$p < 0.0001$. The data underlying S3B Fig can be found in S5 Table Source data.
(TIF)

**S4 Fig. wek knock-down caused DAN loss in the absence of infection.** *wek-RNAi* knock-down caused a decrease in TH + DANs PPL1 **(A)** and PPL2 **(B)**, that was not rescued with the over-expression of *wek* caused by *B. bassiana* infection, suggesting that *wek* is required for DAN differentiation. **(A)** Two-way ANOVA: Infected versus not-infected p = 0.0455; Genotypes: $p = 0.0695$; Interaction: $p = 0.0038$ followed by Tukey's multiple comparisons correction test: non-infected control brains $n = 8$, infected control brains $n = 9$, non-infected *wek-RNAi* brains $n = 7$, infected *wek-RNAi* brains n = 9. (B) Two-way ANOVA: Infected versus not-infected $p = 0.0104$; Genotypes: $p = 0.8564$; Interaction: $p = 0.0217$ followed by Tukey's multiple comparisons correction test: non-infected control brains $n = 8$, infected control brains $n = 9$, non-infected *wek-RNAi* brains n = 7, infected *wek-RNAi* brains n = 9. Graphs show box plots

around the median. Asterisks on graphs: $*p < 0.05$, $**p < 0.01$, $***p < 0.001$, $****p < 0.0001$. The data underlying S4A and S4B Fig can be found in S5 Table Source data.
(TIF)

**S5 Fig. Over-expression of *Toll-1, wek* or *sarm* did not affect climbing. (A)** Over-expression of activated *Toll-1^{10b}* in DANs. OneWay ANOVA: p = 01601, controls: *THGAL4/+; R58E02GAL4/+*. $n = 76$ flies; UAS-Toll-1^{10b}/+: $n = 59$ flies; GAL4/UAS-Toll-1^{10b} $n = 82$ flies. **(B)** Over-expression of *wek* in DANs. OneWay ANOVA: p = 0.3264. controls: *THGAL4/+; R58E02GAL4/+*. $n = 76$; UAS-wek-HA/+: $n = 41$ *THGAL4/+; R58E02GAL4/UAS-wek-HA* $n = 59$ flies. **(C)** Over-expression of *sarm* in DANs. OneWay ANOVA p = 0.1367. Controls: *THGAL4/+; R58E02GAL4/+*. $n = 76$ flies; UAS-sarm/+: $n = 61$ flies; *THGAL4/+; R58E02GAL4/ UAS-dsarm* $n = 60$ flies. Genotypes: **(A)** *THGAL4/UAS-Toll-1^{10b}; R58E02GAL4/+;* **(B)** *THGAL4/+; R58E02GAL4/UAS-wek-HA;* **(C)** *THGAL4/+; R58E02GAL4/UAS-dsarm.* Controls: *THGAL4/+; R58E02GAL4/+* and UAS lines crossed to wild-type (Oregon). The data underlying S5A, S5B and S5C Fig can be found in S5 Table Source data.
(TIF)

**S1 Video. Infected flies at 7 days are mobile.** Climbing assay of one cohort of wild-type flies (Oregon/CantonS), non-infected on the left, infected on the right. Note that all flies are mobile.
(MP4)

**S1 Table. Stock list.** This includes all the Drosophila strains used in this work, their genotypes and origin.
(DOCX)

**S2 Table. List of primers, which were used for qRT-PCRs and molecular cloning.** The table includes the primer name, target gene and primer sequence.
(DOCX)

**S3 Table. List of antibodies.** It includes the names of the primary and secondary antibodies used for immunostaining experiments, the fluorophore linked to secondary antibodies, and the working dilutions.
(DOCX)

**S4 Table. Statistical analysis details, with full genotypes, sample sizes, statistical tests and *p* values.**
(XLSX)

**S5 Table. Source data.** Numerical data used to generate the graphs in each of the figures, shown in separate tabs.
(XLSX)

## Acknowledgments

We are very grateful to Jean Marc Reichhart for advice in the early phases of the project, and to him and Jean-Luc Imler for the gift of *B. bassiana* strains. We thank Anna Parsons and Francisca Rojo-Cortés for constructive criticisms on the manuscript.

## Author contributions

**Conceptualization:** Deepanshu N. D. Singh, Alicia Hidalgo.

**Data curation:** Deepanshu N. D. Singh, Abigail R. E. Roberts, Xiaocui Wang, Guiyi Li, Alicia Hidalgo.

**Formal analysis:** Deepanshu N. D. Singh, Alicia Hidalgo.

**Funding acquisition:** Deepanshu N. D. Singh, Alicia Hidalgo.

**Investigation:** Deepanshu N. D. Singh, Abigail RE Roberts, Xiaocui Wang, Guiyi Li, David Alliband, Elizabeth Ballou, Alicia Hidalgo.

**Methodology:** Deepanshu N. D. Singh, Abigail R. E. Roberts, Enrique Quesada Moraga, Alicia Hidalgo.

**Project administration:** Alicia Hidalgo.

**Resources:** Enrique Quesada Moraga, Alicia Hidalgo.

**Supervision:** Elizabeth Ballou, Hung-Ji Tsai, Alicia Hidalgo.

**Validation:** Deepanshu N. D. Singh, Abigail R. E. Roberts, Xiaocui Wang, Guiyi Li.

**Visualization:** Deepanshu N. D. Singh, Guiyi Li.

**Writing – original draft:** Deepanshu N. D. Singh.

**Writing – review & editing:** Deepanshu N. D. Singh, Abigail R. E. Roberts, Xiaocui Wang, Guiyi Li, Enrique Quesada Moraga, Elizabeth Ballou, Hung-Ji Tsai, Alicia Hidalgo.

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
