## [Editor Report · Decision Letter 0]

14 Aug 2024

Dear Dr Hidalgo, 

Thank you for submitting your manuscript entitled "Fungi activate Toll-1 dependent immune evasion to induce cell loss in the host brain" for consideration as a Research Article by PLOS Biology.

Your manuscript has now been evaluated by the PLOS Biology editorial staff, as well as by an academic editor with relevant expertise, and I am writing to let you know that we would like to send your submission out for external peer review.

However, before we can send you the decision, we need you to complete your submission by providing the metadata that is required for full assessment. To this end, please login to Editorial Manager where you will find the paper in the 'Submissions Needing Revisions' folder on your homepage. Please click 'Revise Submission' from the Action Links and complete all additional questions in the submission questionnaire.

IMPORTANT: after full submission, we will send you the formal decision giving you three months to fullfill your revision plan; please, feel free to tell me now if you think you will need more time for the revision. When the revision is back, we will send it to the original reviewers at Review Commons for their opinion. 

Once your full submission is complete, your paper will undergo a series of checks in preparation for peer review. After your manuscript has passed the checks it will be sent out for review. To provide the metadata for your submission, please Login to Editorial Manager (https://www.editorialmanager.com/pbiology) within two working days, i.e. by Aug 16 2024 11:59PM.

Kind regards,

Melissa

Melissa Vazquez Hernandez, Ph.D.

Associate Editor

PLOS Biology

---

## [Editor Report · Decision Letter 1]

16 Aug 2024

Dear Alicia,

Thank you for your submitting your manuscript "Fungi activate Toll-1 dependent immune evasion to induce cell loss in the host brain" at PLOS Biology. 

As mentioned previously, we expect to receive your revised manuscript within 3 months following your revision plan; we are of course open if you want to follow any of the other experiments suggested by the reviewers. We will then send the revised manuscript to the reviewers from Review Commons. Please email us (plosbiology@plos.org) if you have any questions or concerns, or would like to request an extension. 

**IMPORTANT - SUBMITTING YOUR REVISION**

*Re-submission Checklist*

*Published Peer Review*

*PLOS Data Policy*

*Blot and Gel Data Policy*

Sincerely,

Melissa

Melissa Vazquez Hernandez, Ph.D.

Associate Editor

PLOS Biology

---

## [Decision Letter · Decision Letter 2]

19 Dec 2024

Dear Dr Hidalgo,

Thank you for your patience while we considered your revised manuscript "Fungi activate Toll-1 dependent immune evasion to induce cell loss in the host brain" for publication as a Research Article at PLOS Biology. This revised version of your manuscript has been evaluated by the PLOS Biology editors, the Academic Editor and the original reviewers from Review Commons.

Based on the reviews and on our Academic Editor's assessment of your revision, we are likely to accept this manuscript for publication, provided you satisfactorily address the remaining points raised by the reviewers by adjusting the text specially regarding the impaired locomotion causes. Please also make sure to address the following data and other policy-related requests.

a) We routinely suggest changes to titles to ensure maximum accessibility for a broad, non-specialist readership, and to ensure they reflect the contents of the paper. In this case, we would suggest a minor edit to the title, as follows. Please ensure you change both the manuscript file and the online submission system, as they need to match for final acceptance:

"Toll-1-dependent immune evasion induced by fungal infection leads to cell loss in the Drosophila brain"

b) Please spell the full name Beauveria bassiana in the abstract, in methods and on the first mention in the main text. 

Please supply the numerical values either in the a supplementary file or as a permanent DOI’d deposition for the following figures:

Figure 1BC, 3AEFG, 4BDEHK, 5CEGIJ, 6CEGIJ, 7CEGIJ, 8BD, S3B, S4AB, S5ABC

d) Please cite the location of the data clearly in all relevant main and supplementary Figure legends, e.g. “The data underlying this Figure can be found in S1 Data” or “The data underlying this Figure can be found in https://doi.org/10.5281/zenodo.XXXXX” 

e) Please note that by journal policies, all data should be provided either as Supplementary Material or on a repository such as Zenodo. Anything that is not in the Supplementary Material should be deposited in a stable, community-accepted repositories, or somewhere like zenodo, and the accession numbers provided. For the Microscopy images in Figures 2A-G, 3BC, 4ACFJ, 5ADF, 6ADF, 7ADF, 8AC, S1ABC, S2A-E, S3A, the "image data resource" si a good option, I think https://idr.openmicroscopy.org/about/index.html - information on how this can be done can be found here https://idr.openmicroscopy.org/about/submission.html.

f) Please ensure that your Data Statement in the submission system accurately describes where your data can be found and is in final format, as it will be published as written there.

g) Per journal policy, if you have generated any custom code during the course of this investigation, please make it available without restrictions upon publication. Please ensure that the code is sufficiently well documented and reusable, and that your Data Statement in the Editorial Manager submission system accurately describes where your code can be found.

We expect to receive your revised manuscript within two weeks. 

*Published Peer Review History*

*Press*

Sincerely,

Melissa

Melissa Vazquez Hernandez, Ph.D.

Associate Editor

PLOS Biology

REVIEWERS' COMMENTS:

Reviewer #1

The authors performed additional experiments and carefully answered the questions raised by this reviewer. Beauveria cells entering into fly brain has been solidified and overall quality has been considerably increased in this revision.

This reviewer would like to keep the concern that the impaired locomotion of flies might not be solely due to the B. bassiana-induced neurodegeneration.

Minor comments:

Figure 1C, statistic test has not been shown for day 3 of those infected and non-infected flies. However, it was claimed "No effect was seen after three days exposure (better to be "after exposure for three days") to B. bassiana…" (line 114-116). It is confusing regarding the descriptions of Figure 1C legend: 1) Not to be "3-days-old vs 7-days-old but "days after infection"; p values appear twice: "p=0.0122" won't be "**p<0.01", and "p=0.0048" won't be "***(actually ****shown in figure) p<0.001".

Lines 125 and 127 and elsewhere, change "B. bassiana-GFP were" to "B. bassiana-GFP cells were".

Figure 1 and Figure S1, check out the magenta-type whole brain, stained with DAPI or phalloidin?

Line 183-184, "mtk mRNA was upregulated" or NOT since "ns p=0.0634" is shown on Figure F?

Line 220, and sarm.

Line 570, there is a citation issue here.

— — —

Reviewer #2

The authors describe a role for Toll signaling in detrimental neuronal loss associated with Beauveria bassiana fungal infection in Drosophila melanogaster model. They show that neuronal loss and associated defects in climbing ability (and to some extent in survival from the infection) requires Toll signaling branch via wek/Sarm. The authors have done major work on the manuscript since the previous round of commenting via Review commons, and the conclusions are now backed up with stronger evidence. I'm happy with the manuscript as it is now, below just a few notes I'd like the authors to respond and minor suggestions.

I'm still pondering the choice of controls. I would think that crossing the driver line to the genetic background of the RNAi lines used (available at the stock centers) is the best way to control for the (possible) genetic background effect. For me, the benefit of outcrossing the GAL4/UAS constructs to a "random" wild type strain seem an odd control for experiments where RNAi strains in well characterized backgrounds are used. This said, I do not doubt the validity of the results in this manuscript, considering that they have been confirmed with multiple RNAi strains and wt flies.

Line 254 onwards/Figure 5 I-J. It is said in the text that Toll-1 RNAi increased survival (albeit not significantly) but did rescue the climbing phenotype. To me, this seems to be the opposite: The survival of Toll-1 RNAi flies from infection does not significantly differ from the survival of the uninfected flies of the same genotype (hence rescues it), whereas in the case of climbing, the ability is improved but remains significantly worse than in controls. I find the strong increase in survival of the Toll RNAi flies rather surprising, considering that via using the MyD88>, Toll-1 is likely silenced in most fly tissues, including the fat body, the major organ producing AMPs via Toll and Imd pathways. One could think this would be detrimental in the overall humoral immune response against the fungi, but, based on the data here, clearly is not. This is something I'd like the authors to comment on (also in the manuscript text). (If after specific effects on brain function, one could have opted for a neuron-specific driver).

Line 313 Toll10b misspelled

Lines 1133 and 1166 refer to Table S4, but cannot find such a Table

FM4-64 staining looks quite similar exposed and unexposed brains. What are the bright green areas detected in both?

One could mention also in Figure 3 F&G or in the Figure caption that these are OR flies

— — —

Reviewer #3

In the original submission, the authors showed fairly robust evidence that exposure to B. bassiana fungus leads to loss of brain cell function and that several host immune genes are required for this interaction. While the main conclusions were largely supported, the reviewers had some concerns relating to factors like whether the fungus was confirmed to be within the brain or not. The authors have put impressive effort into the revisions, which addresses all comments. Many new figures have been added and text has been revised as recommended by reviewers. Importantly, they have put in several new fluorescence microscopy figures that show negative controls and others that demonstrate the presence of the fungus in and around the brain, among other new additions. With all major concerns addressed, the manuscript will be a strong addition to the literature.

— — —

---

## [Editor Report · Decision Letter 3]

15 Jan 2025

Dear Alicia,

Thank you for the submission of your revised Research Article "Toll-1-dependent immune evasion induced by fungal infection leads to cell loss in the Drosophila brain" for publication in PLOS Biology. On behalf of my colleagues and the Academic Editor, Aaron P. Mitchell, I am pleased to say that we can in principle accept your manuscript for publication, provided you address any remaining formatting and reporting issues. These will be detailed in an email you should receive within 2-3 business days from our colleagues in the journal operations team; no action is required from you until then. Please note that we will not be able to formally accept your manuscript and schedule it for publication until you have completed any requested changes.

PRESS

Sincerely, 

Melissa

Melissa Vazquez Hernandez, Ph.D., Ph.D.

Associate Editor

PLOS Biology
